



# Uncertainty in aerosol hygroscopicity resulting from semi-volatile organic compounds

Olivia Goulden[1], Matthew Crooks[1], and Paul Connolly[1]

[1]The School of Earth, Atmospheric and Environmental Science, The University of Manchester, Oxford Road, Manchester, M13 9PL

*Correspondence to:* matthew.crooks@manchester.ac.uk

**Abstract.** We present a novel method of exploring the effect of uncertainties in aerosol properties on cloud droplet number using existing cloud droplet activation parameterisations. Aerosol properties of a single involatile particle mode are randomly sampled within an uncertainty range and resulting maximum supersaturations and critical diameters calculated using the cloud droplet activation scheme. Hygroscopicity parameters are subsequently derived and the values of the mean and uncertainty are found to be comparable to experimental observations. A recently proposed cloud droplet activation scheme that includes the effects of co-condensation of semi-volatile organic compounds onto a single lognormal mode of involatile particles is also considered. In addition to the uncertainties associated with the involatile particles, concentrations, volatility distributions and chemical composition of the semi-volatile organic compounds are randomly sampled and hygroscopicity parameters are derived using the cloud droplet activation scheme. The inclusion of semi-volatile organic compounds is found to have a significant effect on the hygroscopicity and contributes a large uncertainty. For non-volatile particles that are effective cloud condensation nuclei, the co-condensation of semi-volatile organic compounds reduces their actual hygroscopicity by approximately 25%. A new concept of an effective hygroscopicity parameter is introduced that can computationally efficiently simulate the effect of semi-volatile organic compounds on cloud droplet number concentration without direct modelling of the organic compounds. These effective hygroscopicities can be as much as a factor of two higher than those of the non-volatile particles onto which the volatile organic compounds condense.

## 1 Introduction

The Earth's weather and climate are both strongly influenced by clouds (Morgan et al., 2010; Ohring and Clapp, 1980). Cloud droplet number concentration and size can have a significant effect on cloud albedo by changing the amount of reflected shortwave radiation and absorbed longwave radiation (Twomey, 1977; McCormick and Ludwig, 1967). In addition, the abundance of cloud droplets and their properties can influence precipitation rate and subsequently cloud lifetime (Stevens and Feingold, 2009; Albrecht, 1989), which itself has a strong interdependency with cloud albedo (Twomey,





1974, 1977). The net global mean radiative forcing is estimated to be reduced by about 0.7 W m$^{-2}$ as a result of aerosol-cloud interactions (Forster et al., 2007). This figure, however, is subject to a large degree of uncertainty.

In general, there is a positive correlation between aerosol number concentration and cloud droplet number concentration (Twomey (1959)), however, the details are much more complex. According to Köhler Theory (Köhler (1936)), the presence of sufficiently large aerosol particles can impede the growth, and subsequent activation, of smaller particles in a polydisperse aerosol by reducing the water available to activate the remaining cloud condensation nuclei (CCN) (Ghan et al. (1998)). Size

and composition are significant in establishing how effectively individual aerosol particles will act as CCN (Pruppacher and Klett (1977)). In addition, the effects of other atmospheric constituents, such as surfactants, can be equally as important in determining cloud droplet number as the number concentration of aerosol particles ((Lance et al., 2004; Nenes et al., 2002)).

A dominant factor influencing aerosol composition is the co-condensation of semi-volatile or-
ganic compounds (SVOCs) onto CCN (Topping and McFiggans (2012)). Köhler Theory is limited to non-volatile compounds, so does not consider the effects of compounds of ranging volatility in the atmosphere. It has been shown that SVOCs increase the tendency for activation of CCN which consequently affects radiative properties of clouds, hence the necessity to quantify their influence (Topping et al. (2013)).

Depending on geographical location, between 5% and 90% of total aerosol mass can be composed of organic material (Andreae and Crutzen, 1997; Zhang et al., 2007; Gray et al., 1986). A portion of this will originate from primary sources but a significant and uncertain amount will be be produced by secondary processes, namely, nucleation of new particles and condensation of SVOCs onto existing particles. The former process increases the number concentration of aerosol particles while the
latter increases the size, and consequently soluble mass, of existing aerosol particles. The enlarged size and altered chemical composition of the particles has a dominant effect on cloud droplet number (Dusek et al., 2006; Topping et al., 2013) and so uncertainties in the amount and composition of secondary organic aerosol mass translate into large uncertainties in cloud properties.

Multiple parameterisations of cloud droplet activation have been developed (Fountoukis and Nenes,
2005; Abdul-Razzak et al., 1998; Abdul-Razzak and Ghan, 2000; Shipway and Abel, 2010; Ming et al., 2005) and have been effective at estimating CCN concentrations at a range of atmospherically applicable conditions (Ghan et al., 2011; Simpson et al., 2014) whilst being more computationally efficient than a detailed cloud parcel model. Although the work of Fountoukis and Nenes (2005) and Abdul-Razzak et al. (1998) has shown to be representative of physical processes (Ghan et al.
(2011)), they lack the consideration of co-condensation of organic vapours.

Connolly et al. (2014) extended the parameterisations of Fountoukis and Nenes (2005) and Abdul-Razzak et al. (1998) to incorporate the effects of co-condensation of SVOCs in the presence of a single non-volatile aerosol mode with lognormally distributed particle sizes. This is achieved by





first assuming the SVOCs are in equilibrium between a vapour and condensed phase at the initial
temperature, pressure and relative humidity; calculated using a molar based equilibrium absorptive
partitioning theory (Barley et al., 2009). The additional mass from the condensed phase of the or-
ganics is added to the non-volatile constituent and the particle size distribution altered so that the
number concentration and geometric standard deviation are the same as the non-volatile mode but
the median diameter is increased to conserve mass. Equilibrium absorptive partitioning theory at
cloud base (99.999%RH) is then used to calculate additional aerosol mass from the organics but
both the median diameter and geometric standard deviation are changed to simulate the condensed
phase of SVOCs after undergoing dynamic condensation during cloud activation. This is carried out
whilst maintaining arithmetic standard deviation and conserving mass. The aerosol size distribution
and material properties at cloud base are then input into the existing cloud droplet activation schemes
of Fountoukis and Nenes (2005) and Abdul-Razzak et al. (1998). The Fountoukis and Nenes (2005)
parameterisation was found to most successfully replicate the results from a detailed parcel model
with binned microphysics and is, consequently, the only parameterisation considered in this paper.
This parameterisation was later extended to include multiple non-volatile aerosol modes (Crooks
and Connolly, 2017).

Petters and Kreidenweis (2007) present the hygroscopicity, $\kappa$, as a method of characterising CCN
activity through relating dry diameter and supersaturation into a single parameter. Typically, for at-
mospheric aerosol the hygroscopicity lies in the range $0.1 < \kappa < 0.9$ with insoluble particles having
a $\kappa$ of approximately zero and $\kappa > 1$ indicating particles that are highly effective as CCN, such as
Sodium Chloride. The hygroscopicity parameter is capable of quantifying water uptake character-
istics for internally mixed particles, and aids in interpreting CCN particles where the composition
is not fully known, by fitting to experimental data. Alternatively, when composition is known, a
volume-averaged mixing rule can be used to determine $\kappa$.

     In the case of involatile particles, the hygroscopicity depends solely on chemical composition and
is independent of particle size. In environments that contain semi-volatile organic compounds, the
hygroscopicity becomes more ambiguous. Due to the condensed mass of semi-volatile organic com-
pounds depends on relative humidity, aerosol particles have chemical compositions and sizes that
vary with the RH. Consequently, the properties of aerosol particles, including the hygroscopicity,
change drastically as they rise in the atmosphere from subsaturated air into cloud. We introduce
three single-parameter measures of the hygroscopicity that incorporate the semi-volatile organic
compounds in different ways. The first, which we denote by $\kappa_{SVOC}$, is calculated using a super-
saturation and CCN concentration calculated in cloud and calculate the critical diameter of the dry
aerosol using the aerosol size distribution at 50% RH. This approach is similar to that used in field
measurements. The second hygroscopicity, $\kappa_{noCC}$, includes semi-volatile organic compounds in the
condensed phase at 90% RH but neglects any further condensation as the humidity rises. This defi-
nition is used to reflect approaches that are currently used in models such as WRF-Chem to include





the partitioning of SVOCs into the particle phase under subsaturated conditions. The third measure, called the effective hygroscopicity and carries a superscript $e$, describes the value of the hygroscopicity of the aerosol particles without co-condensation that is required in order to produce the same number of CCN when co-condensation is included. This method is applied to the aerosol both with

and without the condensed phase of the SVOCs at 90% RH. The effective hygroscopicity could be used in models that currently do not have the capacity to simulate the formation of secondary aerosol mass or co-condensation of SVOCs.

## 2 Methodology for involatile aerosol

There are many sources of uncertainties discussed in this paper and, in order to represent their effect

on cloud droplet formulation, we calculate the hygroscopicity parameter, $\kappa$, introduced by Petters and Kreidenweis (2007), which is defined as

$$\kappa \quad = \quad \frac{4A^3}{27D_d^3 \ln^2 S_c}. \tag{1}$$

Here $D_d$ is the diameter of the dry particle that activates at a percentage supersaturation of $s_c = (S_c - 1) \times 100$, where $S_c$ is the saturation ratio. The parameter $A$ is defined as

$$A \quad = \quad \frac{4\tau M_w}{RT\rho_w},$$

where $\tau$ is the surface tension of water, $M_w$ and $\rho_w$ are the the molecular weight and density of water, and $R$ and $T$ are the universal gas constant and temperature, respectively.

Both the critical diameter and supersaturation are dependent on the chemical composition of the aerosol particles with less hygroscopic particles requiring a larger supersaturation to activate, which

also corresponds to a larger critical diameter. Typically, the critical diameter and supersaturation pairs are obtained from experiments (Svenningsson et al., 2006; Dinar et al., 2006; Petters et al., 2006) but this is a costly and time consuming process. In order to calculate the sensitivity of $\kappa$ to each parameter, a large number of experiments is required. In this paper, a cloud activation parameterisation (Fountoukis and Nenes, 2005) is used to calculate the critical diameter and supersaturation

as a function of the aerosol properties. For each set of parameter values, the parameterisation can calculate the critical supersaturation and number of CCN in under a second. Consequently, this approach offers a practical method to perform a large number of simulations to fully explore the dependence of $\kappa$ on the different model parameters.

In this section we demonstrate how the parameterisations can be used to calculate the uncertainty

in $\kappa$ for common non-volatile compounds before extending the method to include SVOCs in Section 4. The chemical properties of non-volatile compounds can often be measured accurately and so the main source of uncertainty results from measuring the size distribution of the particles. Deviation from ideality is simulated through an uncertainty in the van't Hoff factor. This section demonstrates that the uncertainty in the size distribution that we simulate produces similar uncertainty in $\kappa$ that





are observed in experiments. It also offers a comparison of the uncertainty in $\kappa$ that result from the
inclusion of the SVOCs in Sections 4 and 5.

Particle sizes are assumed to follow a lognormal size distribution of the form

$$\frac{dN}{d\ln D} = \frac{N}{\sqrt{2\pi}\ln\sigma} \exp\left[-\left(\frac{\ln(D/D_m)}{\sqrt{2}\ln\sigma}\right)^2\right], \tag{2}$$

where $N$, $D_m$ and $\ln\sigma$ are the aerosol number concentration, median diameter and geometric stan-
dard deviation, respectively. The cloud droplet activation scheme calculates a maximum supersatu-
ration, $s_{max}$, and a number of CCN, which we denote $N_{CCN}$. We define the critical diameter, $D_d$,
as the smallest diameter of particle that activates, assuming all larger particles also activate. As such,
the critical diameter can be obtained by integrating the size distribution, (2), with respect to $D$ from
$D_d$ up to infinity and equating to the number of CCN calculated by the parameterisation. Therefore,
$D_d$ satisfies

$$\frac{1}{2}N\text{erfc}\left(-\frac{\ln(D_d/D_m)}{\ln\sigma\sqrt{2}}\right) = N_{CCN}, \tag{3}$$

where $\text{erfc}$ is the complementary error function.

In order to encapsulate the uncertainty in the measured size distribution in $\kappa$, we first ran a Monte
Carlo simulation that solves the parameterisation with each size distribution parameter sampled from
normal distributions with specified mean and uncertainty. The range of aerosol size distributions that
this corresponds to is represented by the grey shaded region in the lower plot of Figure 1. After
running the parameterisation, a range of $s_{max}$ and $N_{CCN}$ are obtained; examples of the resulting
probability distributions are shown by the bar charts with blue bars in Figure 1. The mean and
standard deviation of $s_{max}$ and $N_{CCN}$ are calculated to produce approximate normal probability
distributions, shown by the solid black lines. To calculate $\kappa$, a random pair of $s_{max}$ and $N_{CCN}$ were
selected at random from their probability distributions (solid black lines). The value of $N_{CCN}$ and
the mean value of $D_m$ were used to calculate the critical diameter, $D_d$, through equation (3) . This,
together with $s_{max}$, were used to calculate $\kappa$ using equation (1) by setting $s_c = s_{max}$.

When measuring CCN concentrations experimentally, the supersaturation within an instrument is
kept fixed and the resulting CCN concentrations are then measured. Both the supersaturation and
the counting efficiency of the CCN are subject to uncertainty and the normal distributions that are
calculated from the parameterisation are intended to represent this. The uncertainty in supersatura-
tion of a Droplet measurement Technologies dual-column CCN counter (Roberts and Nenes, 2005),
for example, is approximately $\pm 10\%$ (Roberts et al., 2010; Trembath, 2013) and the uncertainty in
counting efficiency is approximately $\pm 6\%$ (Trembath, 2013). The standard deviation in the analo-
gous quantities in our approach are both approximately $\pm 12\%$.

The focus of this paper is on the effect of semi-volatile organic compounds on the hygroscopicity
and, consequently, a thorough analysis of the sensitivity of $\kappa$ to the mean values assigned to the
number concentration, median diameter and geometric standard deviation of the non-volatile particle





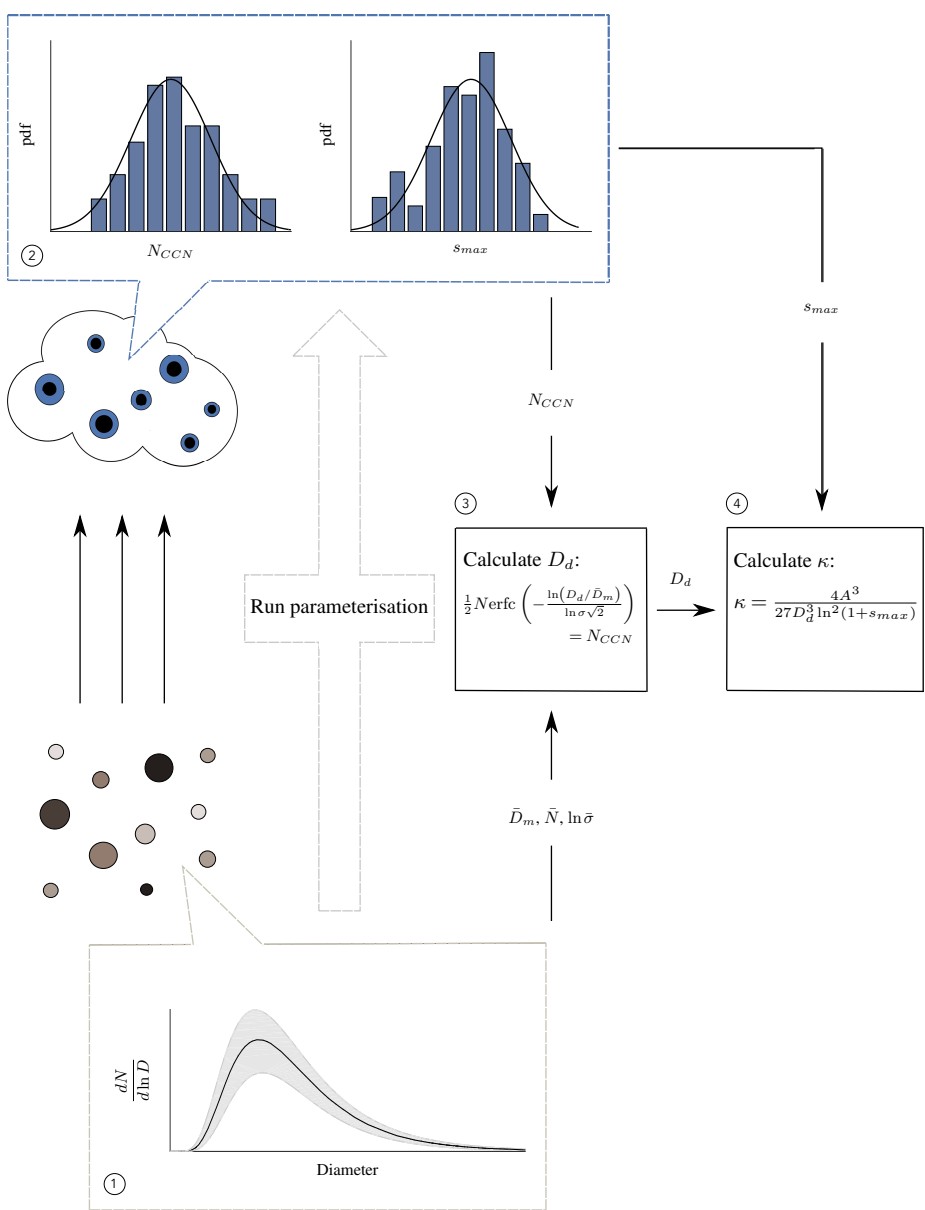

**Figure 1.** A schematic of the method used to calculate the hygroscopicity, $\kappa$, for the non-volatile aerosol particle case. Mean size distribution and uncertainty range is shown in step 1 by the solid black line and grey shaded regions, respectively. Probability distributions of $N_{CCN}$ and $s_{max}$ from the Monte-Carlo simulations are shown by the blue bar charts step 2 and corresponding fitted normal distributions are shown by the overlaid solid black lines. Step 3 combines the mean size distribution with randomly sampled $N_{CCN}$ to calculate a range of $D_d$. In step 4, each $D_d$ is used together with a randomly sampled $s_{max}$ to find the range of $\kappa$.





size distribution is not performed. Plots showing the effects for a few select choices of mean size
distribution parameters are given in the supplementary material. The mean values that are used in the
main body of this paper are given in Table 1. The number concentration was randomly sampled from
a normal distribution with a standard deviation of 10% of the mean value, see Appendix A for further
details. This results in 67% of simulations being run with a number concentration within ±10%
of the mean. The median diameter was also sampled with a normal distribution with a standard
deviation of 10%. The geometric standard deviation was found to vary by 0.1 regardless of the mean
value. As such, the geometric standard deviation was randomly sampled from a normal distribution
with standard deviation equal to 0.1. The uncertainties in each size distribution parameter are also
given in Table 1 as standard deviations of the normal distributions. Ten different logarithmically-
spaced updraft velocities are used ranging from 0.01 m s$^{-1}$ to 10 m s$^{-1}$. Each updraft velocity
produces a different mean value of $S_{max}$ but the resulting hygroscopicity shows little dependence
and for this reason no distinction is made between the different updraft velocities.

**Table 1.** Mean and standard deviation of the size distribution parameters of the non-volatile particles.

| Parameter | mean | standard deviation |
|---|---|---|
| $N$ (cm$^{-3}$) | 1000 | 100 |
| $D_m$ (nm) | 100 | 10 |
| $\ln \sigma$ | 0.5 | 0.1 |

When water condenses onto aerosol particles some or all of the aerosol may dissolve. In addi-
tion, the soluble compounds may dissociate into multiple ions. To simulate the uncertainty in the
solubility and dissociation we randomly sampled the van't Hoff factors from normal distributions
with means, standard deviations and maximum and minimum values stated in Table 2. The maxima
for ammonium sulphate, sodium chloride and sulphuric acid are dictated by ideal behaviour, while
the maximum for levoglucosan is chosen to avoid erroneously high values. The minimum values
of zero are enforced to avoid numerical complications associated with negative van't Hoff factors,
however, this is purely academic as the means and standard deviations are chosen such that it is
highly improbable that the chosen values will lie in this range.

### 3 Results for involatile aerosol

Using the methodology described in Section 2, we produced a range of $\kappa$ for four test compounds;
levoglucosan, ammonium sulphate, sodium chloride and sulphuric acid. Although our method is
more similar to the CCN derived $\kappa$, our results are also compared against the values calculated using
a growth factor. This is due to the lack of experimental uncertainty for the CCN derived $\kappa$. Of our
four test compounds, only levoglucosan has CCN derived mean and uncertainty. For comparison,
Table 3 shows the mean values and uncertainties from both experimental methods, where available.



**Table 2.** Parameters of the normal distributions from which the van't Hoff factors are sampled and the standard deviations are chosen to be 10% of the mean. Randomly sampled values that lie outside of the range of the minimum and maximum are ignored.

| Compound | minimum | mean | maximum | standard deviation |
|---|---|---|---|---|
| Levoglucosan | 0 | 1 | 2 | 0.1 |
| Ammonium Sulphate | 0 | 2.7 | 3 | 0.27 |
| Sodium Chloride | 0 | 2 | 2 | 0.2 |
| Sulphuric Acid | 0 | 3 | 3 | 0.3 |

Our Monte Carlo simulation was run for 1000 different particle size distributions, each of which were run at ten different updrafts, evenly distributed in logspace from 0.01 m s$^{-1}$ to 10 m s$^{-1}$. This produced a range of $N_{CCN}$ and $s_{max}$ for each of the four compounds. Using these values, we were able to calculate a range of critical diameters, $D_d$, using the method described in Section 2. The $D_d$ combined with their corresponding value of $s_{max}$ resulted in a range of $\kappa$, using equation (1). The values of $\kappa$ are restricted to be always positive but, theoretically have no upper bound. As a result, the data is not necessarily normally distributed. Consequently, we have chosen to use the 16.5th and 83.5th quantiles to represent the uncertainty in $\kappa$. This allows for non-symmetric uncertainties but represents the same number of data points that would be represented by the standard deviation if the $\kappa$ values were normally distributed. In the case when the $\kappa$ values are normally distributed, our uncertainty will be equal to the standard deviation of the data.

In Figure 2, the growth factor derived $\kappa$ values from Table 1 are plotted against our calculated hygroscopicity, which will be referred to as $\kappa_{nv}$. The mean values of each compound are displayed by the dots and the horizontal error bars depict the growth factor derived $\kappa_{low}$ and $\kappa_{up}$. The vertical error bars show the 16.5th and 83.5th quantiles of the range of $\kappa$ values from our method and the grey dashed line shows the 1:1 line. The mean CCN derived $\kappa$ and the mean values from our data are shown by the crosses.

The mean values that our method calculate are in excellent agreement with those given in Table 3. In general, our values are slightly higher than the growth factor derived values but are in much better agreement with the CCN derived values. The growth factor derived hygroscopicity for sulphuric acid is noticeably lower than $\kappa_{nv}$ but the CCN derived hygroscopicity is, again, in much better agreement. The error bars from our Monte Carlo simulations are comparable to those from the growth factor derived $\kappa$. In the growth factor derived case, the uncertainty in hygroscopicity for levoglucosan is so small that the error bars are obscured by the dot representing the mean values while the analogous uncertainty in $\kappa_{nv}$ is about 20%, although this still represents a small absolute uncertainty due to the mean value being small. For ammonium sulphate, both methods calculate an uncertainty of approximately 40%. The error bars for sodium chloride represent 20% uncertainty for the growth factor derived $\kappa$ while our approach produces an uncertainty of about 40%. The





error bars in $\kappa_{nv}$ for sulphuric acid are comparable to those for ammonium sulphate and sodium chloride but no growth factor derived uncertainty is available for comparison. In general, our method results in an uncertainty on the order of $20-40\%$ for all compounds and is a result of only varying

the size distribution and van't Hoff factors. In reality, there may be additional chemical variations, such as surface tension, that may influence the hygroscopicity. Although we have chosen to use an uncertainty in the van't Hoff factor of 10%, it is possible that the range of values do not accurately capture the deviation from ideality. Our simulations, however, indicate that the van't Hoff factor has little influence on the uncertainty in hygroscopicity compared to variations in the size distribution

(not shown).

**Table 3.** Growth factor and CCN derived mean and uncertainties in $\kappa$ taken from Petters and Kreidenweis (2007).

| Compound | Growth factor | | | CCN | | |
|---|---|---|---|---|---|---|
| | $\kappa_{\mathrm{low}}$ | $\kappa_{\mathrm{mean}}$ | $\kappa_{\mathrm{up}}$ | $\kappa_{\mathrm{low}}$ | $\kappa_{\mathrm{mean}}$ | $\kappa_{\mathrm{up}}$ |
| Levoglucosan | 0.15 | 0.165 | 0.18 | 0.193 | 0.208 | 0.223 |
| Ammonium Sulphate | 0.33 | 0.53 | 0.72 | N/A | 0.61 | N/A |
| Sodium Chloride | 0.91 | 1.12 | 1.33 | N/A | 1.28 | N/A |
| Sulphuric Acid | N/A | 1.19 | N/A | N/A | 0.9 | N/A |

## 4    Methodology including the effects of SVOCs

Calculating $\kappa$ in the presence of SVOCs is highly sensitive to the method that is used. In field measurements, atmospheric aerosol is passed through instruments under subsaturated conditions in order to measure the size distribution and composition while the number of CCN is calculated

under supersaturated conditions. Due to the partitioning of SVOCs between a vapour and condensed phase being sensitive to the relative humidity, the two measurements will include different quantities of SVOCs in the particle phase as some of the higher volatility SVOCs evaporate as the relative humidity is reduced. This results in uncertainty in how much secondary aerosol mass results from the SVOCs but, additionally, in the total abundance of SVOCs that exists, which will influence cloud

droplet activation.

The volatile nature of SVOCs results in new pathways through which the SVOCs affect $\kappa$ that are not present in the non-volatile particle case. The CCN-based approach to calculating $\kappa$ requires integrating the aerosol size distribution to find the diameter, $D_d$, above which particles activate. The aerosol size distribution, in this case, is measured at subsaturated conditions, typically $\approx 50\%$ RH

(Taylor et al., 2016). The number of particles that activate, however, is controlled by the chemical composition and size of the aerosol under supersaturated conditions. This is also true of the maximum supersaturation, $s_{max}$, that is calculated in the parameterisation. As such, the hygroscopicity of

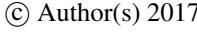



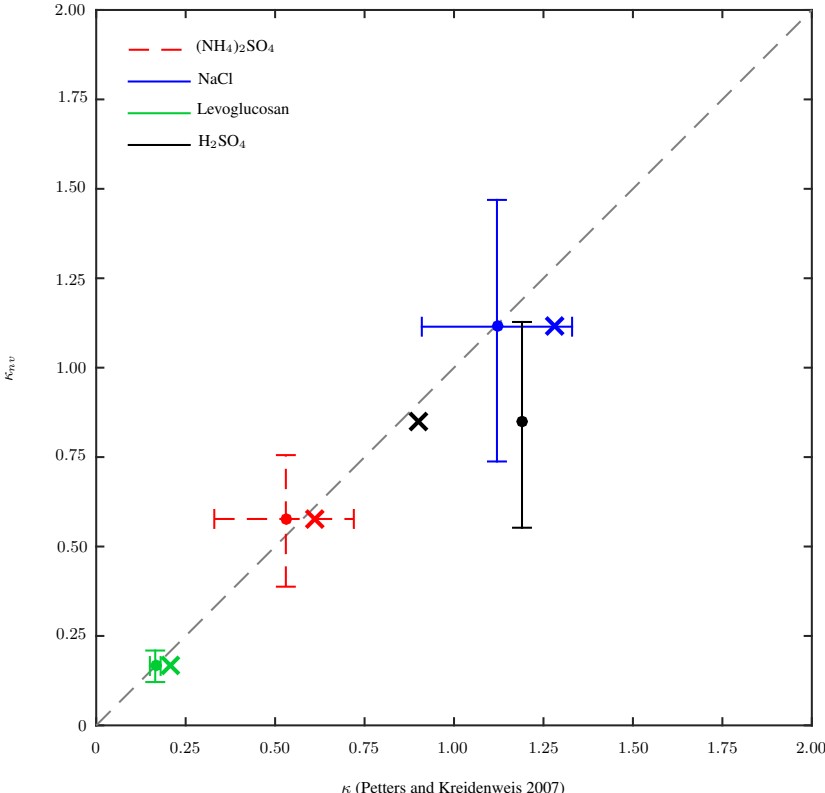

**Figure 2.** Growth factor derived hygroscopicity, $\kappa$, of non-volatile compounds from Petters and Kreidenweis (2007) plotted against the values from our Monte Carlo approach. Mean values from the CCN derived and growth factor derived experimental methods are shown by the crosses and dots, respectively, with $y$ coordinates showing the mean from our approach. Horizontal error bars show $\kappa_{low}$ and $\kappa_{up}$ from the growth factor derived $\kappa$ given in Table 3, and the vertical error bars show the 16.5th and 83.5th quantiles of our derived values. Ammonium sulphate is shown in red with dashed error bars, sodium chloride is shown in blue, levoglucosan is shown by the green and sulphuric acid is shown in black.





the aerosol in the presence of SVOCs cannot simply be related to the chemical composition through the mixing of Petters and Kreidenweis (2007),

$$\kappa = \sum_i \epsilon_i \kappa_i, \tag{4}$$

which is used for internally mixed non-volatile aerosol. Here, $\kappa_i$ is the hygroscopicity of the $i^{th}$ compound and $\epsilon_i$ is the volume fraction of the aerosol occupied by that compound. In the case of ammonium sulphate, sodium chloride and sulphuric acid, adding non-volatile organic compounds with a hygroscopicity of $\approx 0.15$, typical of secondary organic aerosol (Varutbangkul et al., 2006; Massoli et al., 2010), will decrease the hygroscopicity of the aerosol through the relation (4). Since the $\kappa$ of levoglucosan is similar to the hygroscopicity of the organic compounds we consider, the mixing rule will have little effect on the hygroscopicity of the internal mixture.

In the case of semi-volatile organic compounds, as considered in this paper, the condensed mass of SVOCs at 50% RH is used to increase the diameters of the particles in the aerosol size distribution. The additional condensed mass that partitions into the particle phase at cloud base will change the size of the particles but also their affinity to activate into cloud drops in the parameterisation using the mixing rule, (4). To calculate the hygroscopicity including the SVOCs, $\kappa_{SVOC}$, we use the aerosol particles at 100% RH to calculate $N_{CCN}$ and $s_{max}$ but integrate the aerosol size distribution at 50% to obtain $D_d$. These are combined to calculate $\kappa_{SVOC}$ through (1).

Semi-volatile organic compounds are subject to large uncertainties. These arise from the measurements of the saturation vapour pressure, the abundance and material properties; all of which are difficult to measure due to the volatile nature of the compounds. In addition, atmospheric VOCs originate from unknown sources and subsequently undergo an unknown series of gas-phase reactions. The result is a mix of, largely, unidentifiable compounds with unknown chemical properties. In our model, we use the $\log_{10}$ volatility basis set (VBS) of Donahue et al. (2006) with saturation concentrations, $C^*$, ranging from $1 \times 10^{-6}$ $\mu$g m$^{-3}$ to $1 \times 10^3$ $\mu$g m$^{-3}$. Each volatility bin represents multiple organic species with unknown material properties. In our Monte Carlo simulations we randomly select material properties of each volatility bin using a normal distribution with means and standard deviations given in Table 5. The origins of these values are given in Appendix B2 and are based on data in the literature. To simulate uncertainty in the saturation vapour pressures of the individual compounds that are represented by the volatility distribution, we initially begin with the volatility distribution given in Cappa and Jimenez (2010), which is restated in Table 4. Some of the mass concentration in each volatility bin is then randomly redistributed between neighbouring bins to simulate uncertainties in the $C^*$ values of individual compounds. This process adds a random element to the relative mass concentrations in each volatility bin. The total concentration of SVOCs is then random chosen from a uniform distribution so that the bulk organic mass fraction of the aerosol at 50% RH without its associated water is between 0.1 and 0.5. Further details on all of the simu-





lated uncertainties of the SVOCs are given in Appendix B. The size distribution of the non-volatile particles was sampled using the same uncertainties specified in Section 2.

**Table 4.** Volatility distribution of SVOCs from Cappa and Jimenez (2010).

| $\log \mathcal{C}^*$ | -6 | -5 | -4 | -3 | -2 | -1 | 0 | 1 | 2 | 3 |
|---|---|---|---|---|---|---|---|---|---|---|
| $\mathcal{C}_j$ ($\mu$g m$^{-3}$) | 0.005 | 0.01 | 0.02 | 0.03 | 0.06 | 0.08 | 0.16 | 0.3 | 0.42 | 0.8 |

**Table 5.** The range of effective material parameters used for the compounds in each volatility bin. Minimum and maximum values are stated as well as the mean and standard deviation of the normal distribution from which values are sampled.

| Parameter | minimum | maximum | mean | standard deviation |
|---|---|---|---|---|
| Molecular weight (g mol$^{-1}$) | 100 | 300 | 200 | 100 |
| Density (kg m$^{-3}$) | 1000 | 1500 | 1250 | 250 |
| van't Hoff factor | 0 | 1 | 0.5 | 0.5 |

A Monte Carlo simulation was carried out that calculated the range of $s_{max}$ and $N_{CCN}$ that results from the parameterisation of Connolly et al. (2014) when the volatility distributions were randomly chosen from the distributions described above. The aerosol size distributions of the non-volatile particles were randomly chosen in the same way as in Section 2. In each simulation, a set of non-volatile size distribution parameters and a volatility distribution and material properties of

the SVOCs were chosen randomly. Dry aerosol size distributions including condensed SVOCs were calculated at the initial RH of 50% and $D_d$ calculated using a randomly chosen $N_{CCN}$ from within its uncertainty range. The corresponding $s_{max}$ was then used to calculate $\kappa$. A schematic of this process is shown in Figure 3.

We first investigate the effect of uncertainty in these measurements on $\kappa$ that may result from

using a CCN-based method, common in insitu measurements. In Section 5.1, we investigate the dependence of the hygroscopicity on the relative humidity at which the aerosol size distribution is measured. We then go on to calculate effective $\kappa$ values that the non-volatile compounds in the absence of SVOCs would have to have in order to simulate the same cloud droplet activation affinity that the SVOCs induce. This presents a possible computationally efficient method to include the

effect of SVOCs on cloud in large-scale models that currently do not have the capacity to do so directly. The results from this work are presented in Section 5.2.



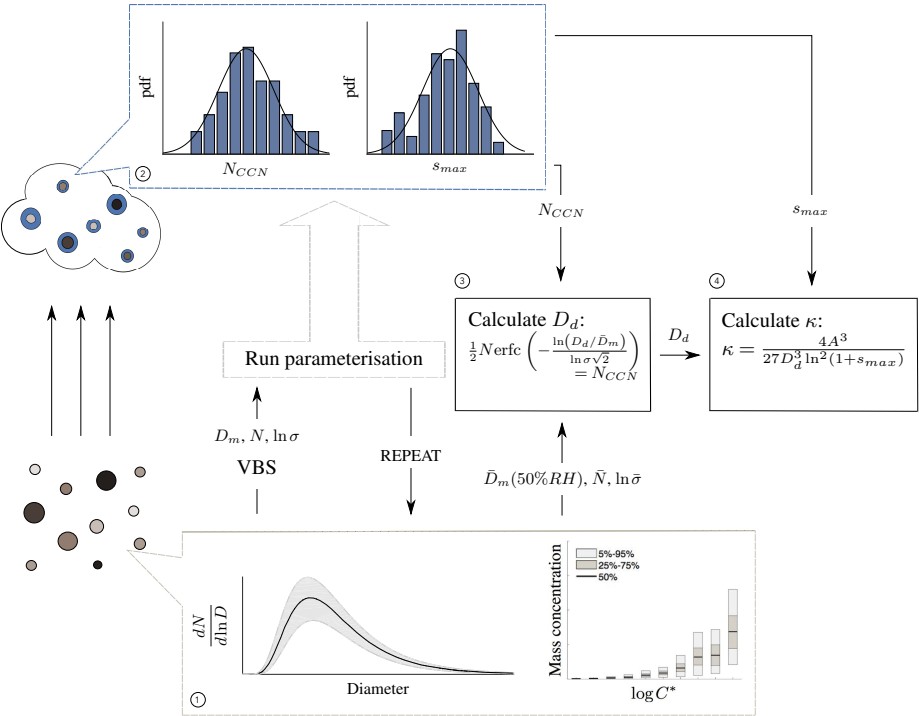

**Figure 3.** A schematic of the method used to calculate the hygroscopicity including the effects of SVOCs. The methodology is the same as in the non-volatile case but with the addition of randomly choosing a volatility distribution together with the non-volatile size distribution parameters as inputs into the parameterisation (step 1). The median volatility distribution is used to calculate the median diameter at 50% RH that is integrated to calculate $D_d$ in step 3.

## 5 Results including the effects of SVOCs

Using the methods detailed in Section 4, we calculated a range of hygroscopicities, $\kappa_{SVOC}$, that incorporate the full effect of SVOCs on ammonium sulphate, sodium chloride, levoglucosan and sulphuric acid with 10%-50% of the total aerosol mass being composed of SVOCs. These are plotted against the hygroscopicities calculated for just the non-volatile modes, $\kappa_{nv}$, from Section 3 in Figure 4. Similarly to previous figures, the error bars show the 16.5th and 83.5th quantile in order to contain the middle 67% of values. The error bars show a marginally smaller uncertainty for $\kappa_{SVOC}$, being roughly 30% of the mean, which is, at most, 10% smaller than the uncertainty associated with the non-volatile particles only.

The influence of SVOCs reduces the hygroscopicity of ammonium sulphate, sodium chloride and sulphuric acid. The hygroscopicity of levoglucosan is largely unchanged, with only a very slight



increase, due to its chemical properties being very similar to that of the SVOCs and, consequently, the mixing rule creates little difference between $\kappa_{nv}$ and $\kappa_{SVOC}$. The more hygroscopic compounds, by comparison, will be more heavily affected because of a larger difference in $\kappa$ between the non-volatile aerosol and the semi-volatile organic compounds.

We also calculated a range of hygroscopicities that include the initial condensed mass of organic vapours at 0% RH but do not consider any co-condensation that would occur during ascent to cloud base. We denote this as $\kappa_{noCC}$. In the absence of co-condensation of SVOCs, the aerosol particles are composed of the non-volatile constituent plus the condensed organic mass at 50% RH and are, subsequently, assumed to be non-volatile in the parameterisation. The resulting $\kappa_{noCC}$ can be calculated using the mixing rule (4) and due to the low hygroscopicity of the SVOCs will be lower than $\kappa_{nv}$. Figure 5 shows that the result is nearly a 30% decrease in mean hygroscopicity in ammonium sulphate, sodium chloride and sulphuric acid with a 15% reduction in levoglucosan.

Figure 6 shows $\kappa_{SVOC}$ plotted against $\kappa_{noCC}$. Both hygroscopicities are calculated using the same aerosol size distribution at 50% RH and the difference is that $\kappa_{SVOC}$ has further condensed mass of SVOCs added before activation. Due to the substantial decrease in hygroscopicity due to the condensed SVOCs at 50% RH, the mixing rule, when applied at cloud base, has a much less significant effect on the aerosol composition. The additional mass, however, will act to increase the diameter of the particles that activate in the parameterisation and this will increase $N_{CCN}$ and, consequently, $\kappa_{SVOC}$. The result is that $\kappa_{SVOC}$ is larger than $\kappa_{noCC}$ for all compounds due to the enhancement in size dominating the change in composition due to co-condensation. Figure 6 shows that the hygroscopicity of ammonium sulphate, sodium chloride and sulphuric acid increase by about 15% due to co-condensation and levoglucosan by 30%.

The difference between $\kappa_{SVOC}$ and $\kappa_{nv}$ in Figure 4 is the combination of both the suppression seen in Figure 5 and the enhancement in Figure 6. For ammonium sulphate, sodium chloride and sulphuric acid the decrease due to the mixing rule at 50% RH is larger in magnitude than the enhancement due to co-condensation and the result is a net decrease of approximately 30%. For levoglucosan, the mixing rule has a less dominant effect than the increase in size at cloud base and, hence, the slight increase in hygroscopicity when co-condensation of SVOCs is included compared to the non-volatile compounds alone.

### 5.1 Dependence of $\kappa$ on relative humidity

Under the supersaturated conditions experienced in clouds, the majority of the SVOCs partition into the condensed phase. This increases the particle sizes and influences the activation process. A significant portion of the higher volatility SVOCs only condense at relative humidities at or above 100%, however, and exist in the vapour phase at lower RH. This means that they influence $\kappa$ through the maximum supersaturation attained and the number of cloud droplets that result. The lower volatility SVOCs are almost always in the condensed phase and have little sensitivity to the relative humidity.



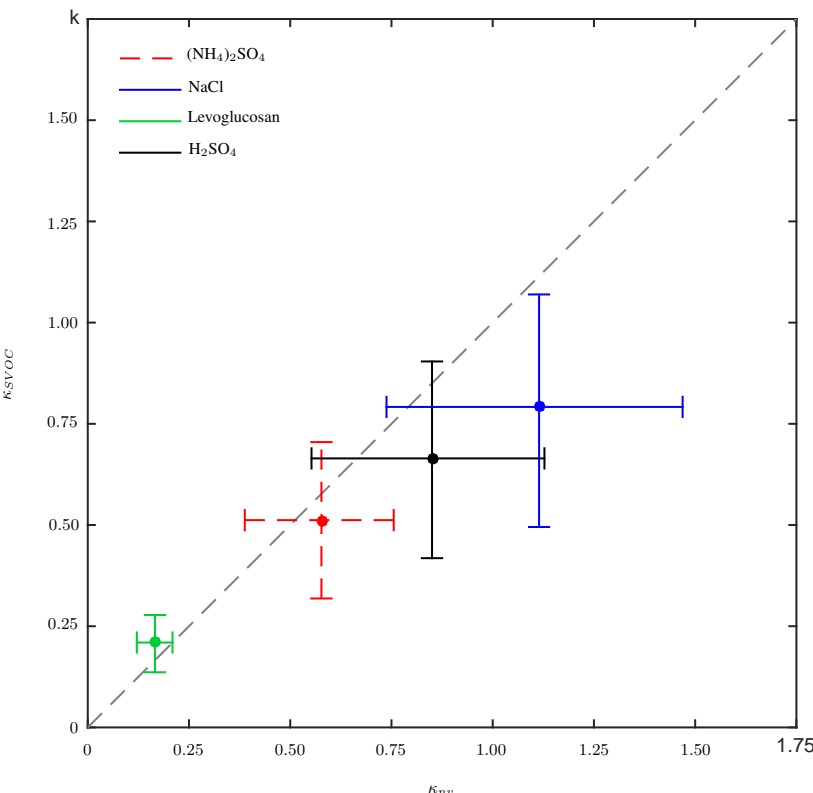

**Figure 4.** Hygroscopicity including the SVOCs, $\kappa_{SVOC}$, plotted against $\kappa_{nv}$. The grey dashed line shows equality between the axes, error bars showing the middle 67% of our derived values and the intersections of the error bars depicting the means. Ammonium sulphate is shown in red with dashed error bars, sodium chloride is shown in blue, levoglucosan is shown by the green and sulphuric acid is shown in black.

In contrast, the mid-volatility SVOCs are more sensitive to lower relative humidity values; partition-

ing differently at RH values between 50% and 90%. This range is typical for the relative humidity that is used when sizing aerosol particles and the concentration of these mid-volatility compounds in the condensed phase can vary significantly. The overall effect on the total condensed organic mass and consequently $D_d$ and $\kappa$, however, is minimal due to the lower volatility compounds always existing in the condensed phase.

To investigate the effect of relative humidity on $\kappa$, we ran the Monte Carlo simulation to calculate the uncertainty in supersaturation and number of CCN but have integrated the aerosol size distribution at different relative humidities in order to calculate different $D_d$. At each relative humidity, the median diameter of the aerosol size distribution was calculated assuming constant geometric standard deviation and conservation of mass. At 99.999% RH only, the aerosol size distribution from the





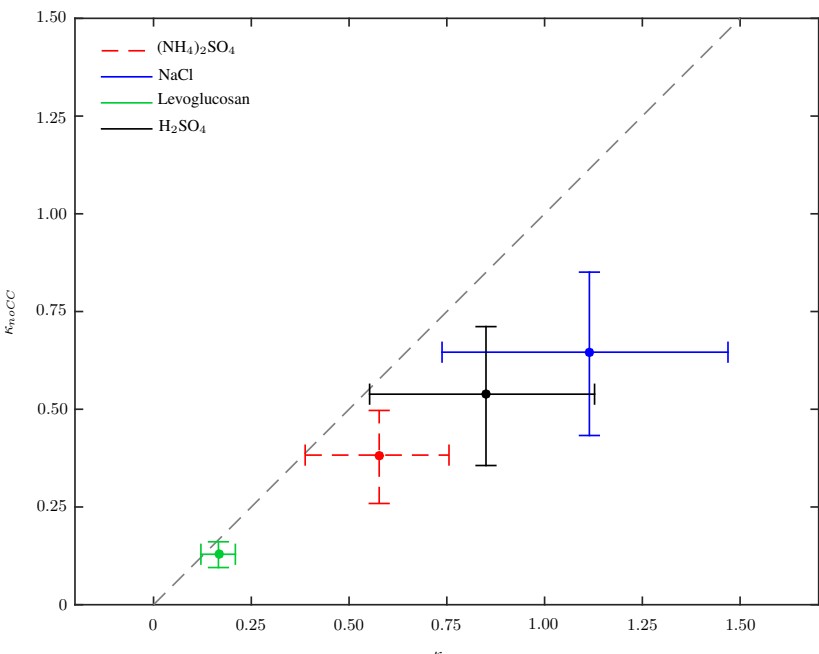

**Figure 5.** Hygroscopicity, $\kappa_{noCC}$, plotted against $\kappa_{nv}$. The grey dashed line shows equality between the axes, error bars showing the middle 67% of our derived values and the intersections of the error bars depicting the means. Ammonium sulphate is shown in red with dashed error bars, sodium chloride is shown in blue, levoglucosan is shown by the green and sulphuric acid is shown in black.

parameterisation at cloud base is used (Connolly et al., 2014). In this case, the geometric standard deviation of the composite aerosol size distribution was recalculated along with the median diameter to simulate the dynamic condensation of the SVOCs during cloud activation.

Figure 7 shows the variation in the $\kappa$ values that results from calculating the dry aerosol size distribution at different initial RH values. The results show a modest dependence on RH with a

tendency for lower $\kappa$ values at higher RH. Mean $\kappa$ values decrease by about 0.07 from an RH of 70% to 95%, however, this decrease is dwarfed by the uncertainty range in $\kappa$ and contributes no significant variation. Due to the different method that is employed to calculate the aerosol size distribution at 99.999% RH, there is a significant difference in the hygroscopicity calculated with a reduction of about 0.2 compared to the lower RH values. This is a result of the narrower size distribution that

results from altering the geometric standard deviation as well as the median diameter.

Figure 8 shows the dependence of the condensed mass of SVOCs, calculated using equilibrium absorptive partitioning theory, on RH for a typical volatility distribution used in this paper. As can be seen, there is little difference in condensed mass between 70% and 80% RH. Some increase in condensed mass occurs at 90% and 95% but it is not until the RH approaches 100% that there





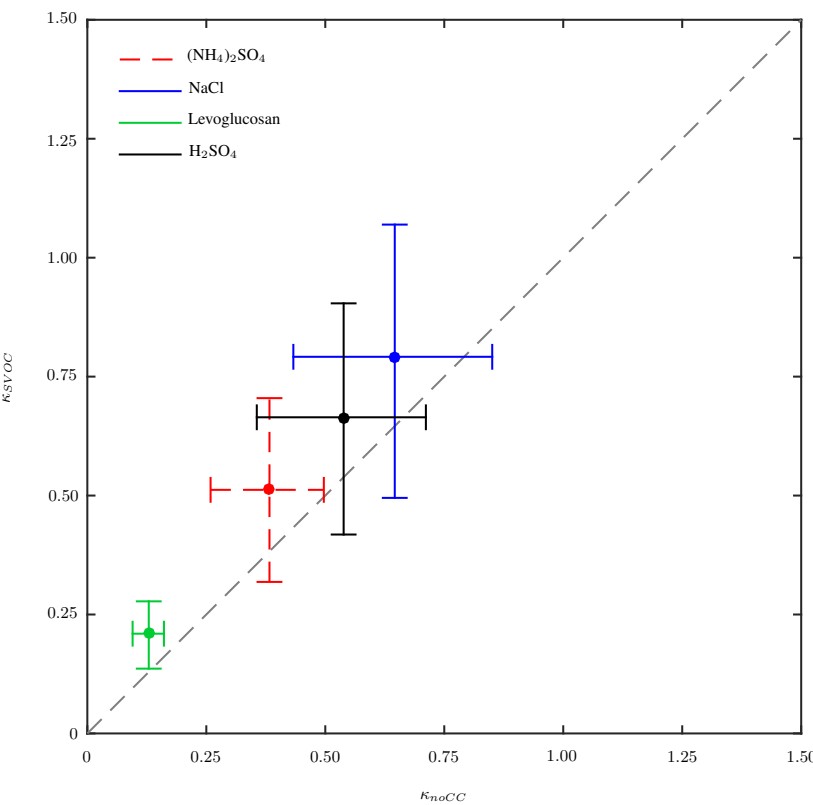

**Figure 6.** Hygroscopicity, $\kappa_{SVOC}$ plotted against $\kappa_{noCC}$. The grey dashed line shows equality between the axes, error bars showing the middle 67% of our derived values and the intersections of the error bars depicting the means. Ammonium sulphate is shown in red with dashed error bars, sodium chloride is shown in blue, levoglucosan is shown by the green and sulphuric acid is shown in black.

is a significant difference in the higher volatility bins. Due to the presence of the lower volatility bins, all of which exist in the condensed phase at 70% RH, the increase in condensed mass in the higher volatility bins contributes little to the total SVOC concentration in the condensed phase below 95% RH. With the addition of the non-volatile aerosol mass, the total aerosol mass changes very little between 70% and 95% RH. A more noticeable dependence may be observed for higher

concentrations of SVOCs, that represent a more significant portion of the total aerosol mass, as well as environments that do not contain lower volatility compounds. Overall, however, it is unlikely that the relative humidity at which the size distribution is measured will have a significant effect on $\kappa$.



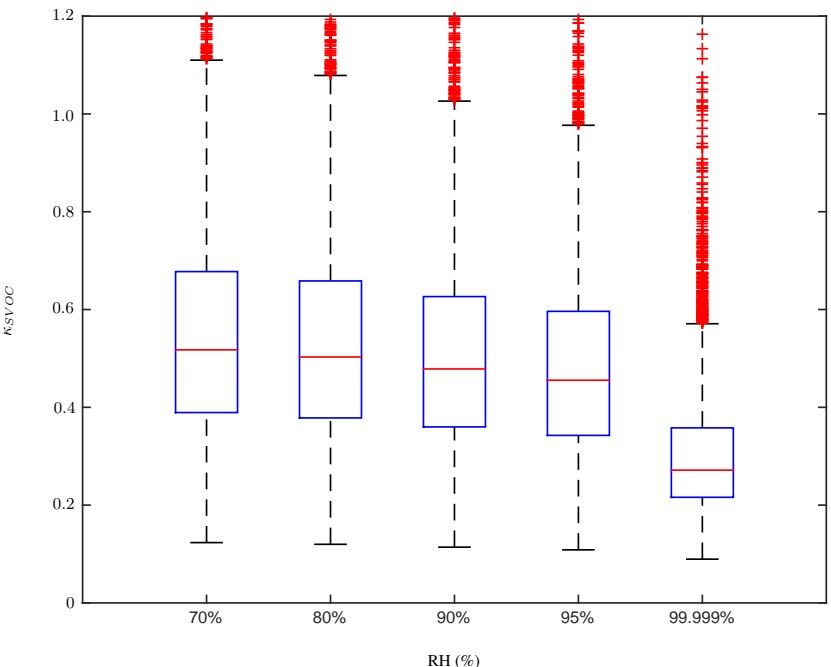

**Figure 7.** Box and whisker plots showing the range of $\kappa$ values that result from the uncertainties in the model inputs using values of $D_d$ calculated at different initial RH values, as specified on the $x$ axis. Each box and whisker represents 50 calculations using ammonium sulphate aerosol

## 5.2 Effective kappa

Including the production, condensation, evaporation, reaction and oxidation of SVOCs directly in
large-scale models is very computationally expensive and is rarely carried out, especially for more than one compound. For the purposes of aerosol transport, it is common to apply equilibrium absorptive partitioning theory to calculate the particle phase of volatile compounds (Tsigaridis et al., 2014). Comprehensively including the effect of SVOCs on cloud droplet activation in large-scale models is yet to be carried out (Ervens, 2015). Although, a number of studies have attempted to parameterise
the relation between secondary organic aerosol and cloud liquid water content (Myriokefalitakis et al., 2011; Lin et al., 2014) an effective cloud droplet radius has to be assumed in order to incorporate them into large-scale models.

    We suggest a potential method to include the effects of SVOCs on cloud droplet activation in large-scale models that is computationally efficient and does include the process of co-condensation as the
relative humidity exceeds 100%. Our method, additionally, allows for a dependence on aerosol properties rather than assuming an arbitrary effective radius of the cloud droplets. Our method involves using our Monte Carlo simulations using the cloud droplet activation parameterisation including the





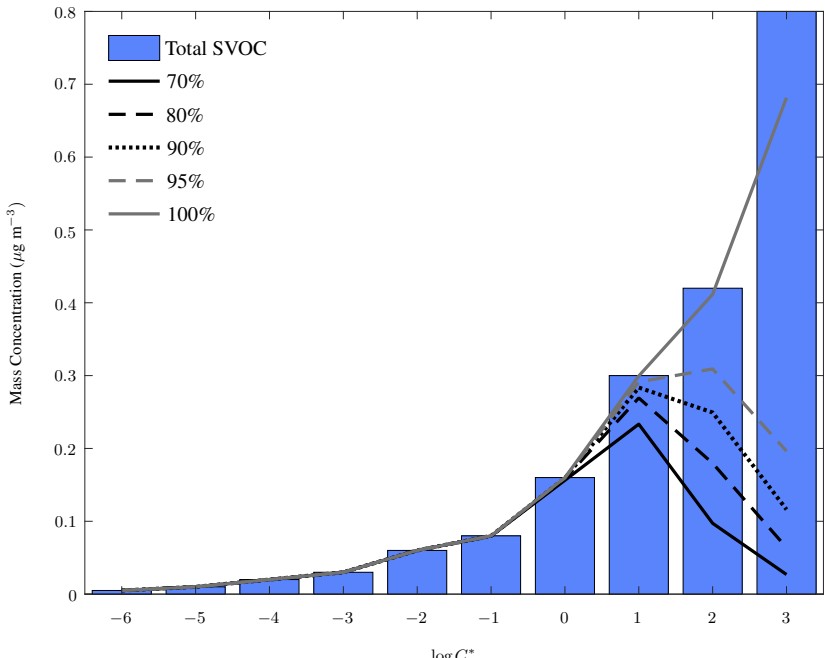

**Figure 8.** Variation in equilibrium condensed mass of SVOCs at different RH. Total concentrations of the SVOCs are shown by the bars while the condensed concentrations are shown by the lines; the corresponding RH values are stated in the legend.

effects of SVOCs (Connolly et al., 2014) to calculate the number of CCN for a given size distribution of non-volatile particles and volatility distribution, together with their associated uncertainties. We

then iterate the parameterisation without SVOCs (Fountoukis and Nenes, 2005) to find the $\kappa$ of the non-volatile particles that would be required in order to produce the same number of cloud droplets that was calculated in the presence of SVOCs. The hygroscopicity of the non-volatile particles is referred to as the effective hygroscopicity and is denoted $\kappa_{nv}^e$. At very low updrafts, the parameterisation can be insensitive to the hygroscopicity and, consequently, there may not exist a value of

$\kappa_{nv}^e$ that produces the same concentration of CCN as in the SVOC case. Similarly, at high updrafts, the parameterisation often activates all particles, even when the hygroscopicity is very low. To avoid these complications, we iterate the vertical updraft in the parameterisation with SVOCs until 90% of particles activate. The parameterisation without SVOCs is then evaluated at this updraft while iterating the hygroscopicity until 90% of particles activate. The resulting hygroscopicity is defined

as the $\kappa_{nv}^e$.

Figure 9 compares the $\kappa_{nv}$ values from our Monte Carlo simulations for just the non-volatile aerosol against the effective $\kappa_{nv}^e$ values of the non-volatile aerosol. The mean effective $\kappa_{nv}^e$ values





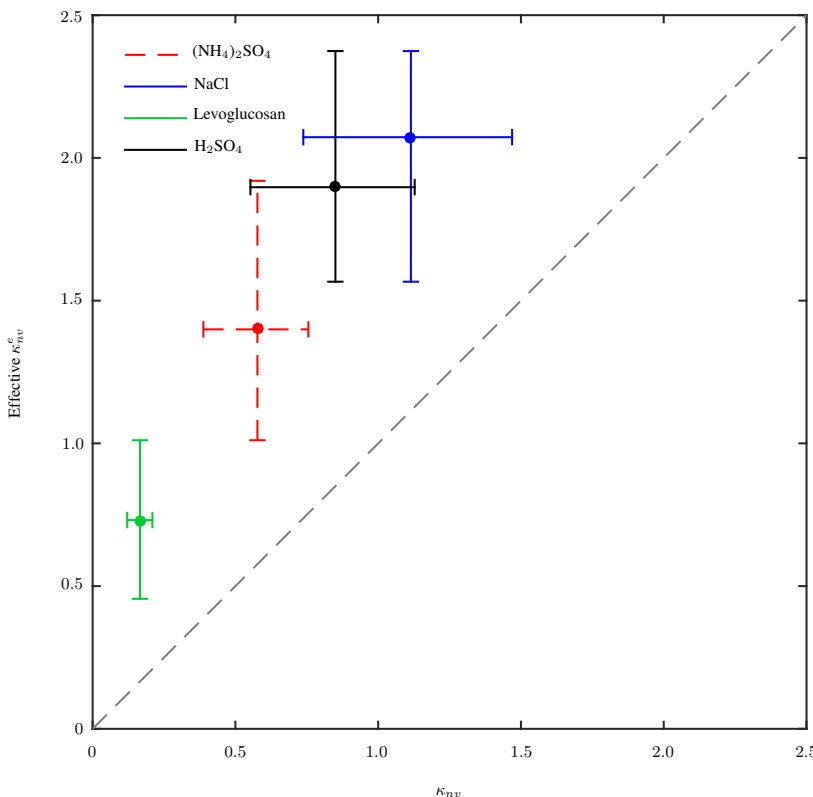

**Figure 9.** Effective hygroscopicity, $\kappa_{nv}^e$ plotted against the hygroscopicity of non volatile compounds, $\kappa_{nv}$. The dots show the mean values and the error bars represent the 16.5th and 83.5th quantiles. Ammonium sulphate is shown in red with dashed error bars, sodium chloride is shown in blue, levoglucosan is shown by the green and sulphuric acid is shown in black. The dashed grey line shows the 1:1 line.

are significantly higher than the mean $\kappa_{nv}$ of the non-volatile compounds in all four cases with an increase of 60% for sodium chloride and a factor of 2 in the case of levoglucosan. The reason for

this increase is that the non-volatile particles are much smaller than the particles that activate in the presence of SVOCs and, consequently, must have a larger $\kappa_{nv}^e$ in order to compensate. An increase in the hygroscopicity of aerosol particles on this scale could have a significant effect in large-scale models. The uncertainty ranges are slightly larger in the effective $\kappa_{nv}^e$ cases than the analogous $\kappa_{nv}$ values but a significant increase is only observed for levoglucosan. This is due to a very small

uncertainty in $\kappa_{nv}$ for levoglucosan rather than a large uncertainty in the effective hygroscopicity. On average, the uncertainty spans about 30-40% of the mean.

     For models that include the condensed mass of SVOCs but neglect the vapour phase during activation, similar calculations can be carried out. The effective hygroscopicity of the aerosol size




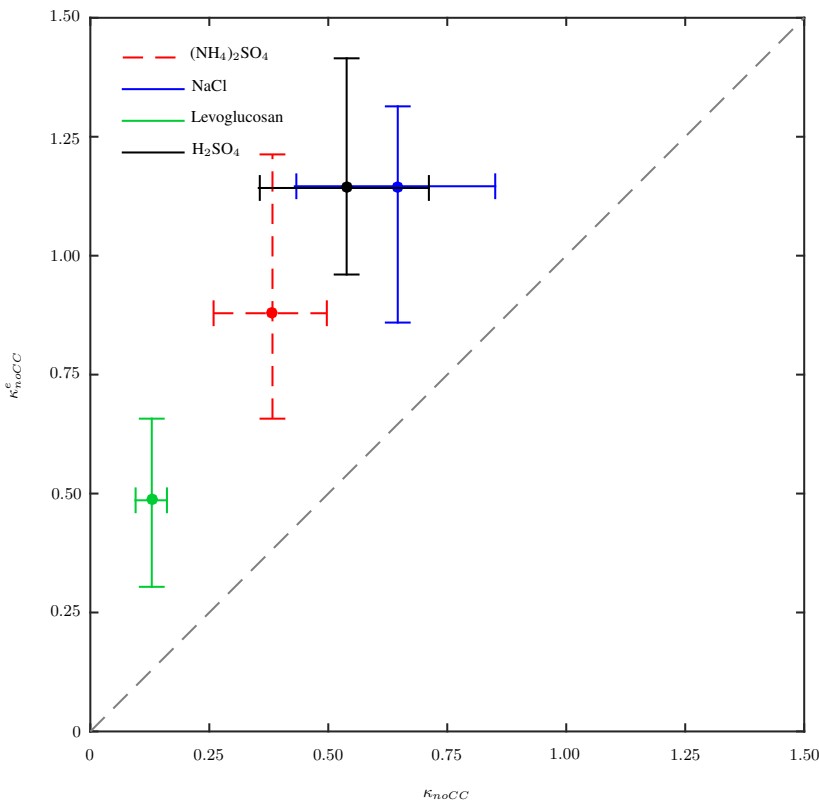

**Figure 10.** Effective hygroscopicity, $\kappa_{noCC}^e$, values plotted against the hygroscopicity including the condensed SVOCs at 50% RH in the absence of further co-condensation, $\kappa_{noCC}$. Error bars represent the 16.5th and 83.5th quantiles. Ammonium sulphate is shown in red with dashed error bars, sodium chloride is shown in blue, levoglucosan is shown by the green and sulphuric acid is shown in black.

distribution at 50% RH, $\kappa_{noCC}^e$, in this case can then be calculated in an analogous way to the non-

volatile case. Figure 10 compares the hygroscopicity of the internally mixed aerosol at 50% RH in the absence of further co-condensation of SVOCs, $\kappa_{noCC}$, against the effective hygroscopicity of such particles, $\kappa_{noCC}^e$. As in the involatile case, the effective $\kappa_{noCC}^e$ values are larger than those without co-condensation, however, the increase is slightly less at only 40% for ammonium sulphate, sodium chloride and sulphuric acid. Levoglucosan increases by a factor of 2, as before. The uncer-

tainty associated with the effective $\kappa_{noCC}^e$ in this case is much smaller than in Figure 9, varying by only about 0.2-0.3, although this still represents a similar 30-40% of the mean value.





## 6 Conclusions

We propose that semi volatile organic compounds have a significant impact on the hygroscopicity of atmospheric aerosol, and therefore the ability for these aerosols to form cloud droplets. The effects of
SVOCs can both increase and decrease $\kappa$. The mixing rule detailed in section 5 can shift $\kappa$ either way, dependent on the hygroscopicity of the non-volatile aerosol at 50% RH. If the hygroscopicity of the non-volatile aerosol is lower than the mean hygroscopicity of the SVOCs then $\kappa$ will be increased, as is the case with levoglucosan. For non-volatile aerosol with higher hygroscopicity than the SVOCs, the mixing rule will have a tendency to decrease $\kappa$ when the SVOCs are included. The magnitude of
the shift in $\kappa$ is dependent upon the difference between the $\kappa$ of the SVOCs and $\kappa$ of the aerosol along with the mass of SVOCs present. Semi-volatile compounds also affect $\kappa$ by enhancing the size of swollen aerosol in the atmosphere, which consequently increases $N_{CCN}$, resulting in a smaller $D_d$ that then produces an increase in $\kappa$. These two effects contrast each other, with one being dominant over the other and which is dominant is dependant on the situation. With a full consideration of
SVOCs, the overall effect is that non-volatile aerosol particles with $\kappa$ greater than that of the SVOCs give a $\kappa_{SVOC}$ which is less than $\kappa_{nv}$, whereas if $\kappa$ of the non-volatile aerosol is smaller than the SVOCs then $\kappa_{SVOC} > \kappa_{nv}$. A larger disparity between the hygroscopicity of the aerosol and the semi-volatile compounds causes a larger translation of the mean $\kappa$.

The effects of SVOCs is also subject to the complexity of their inclusion in the model. We have
shown that by omitting the effects of semi-volatile compounds during co-condensation, we obtain a lower hygroscopicity values than if co-condensation is fully included since $\kappa_{noCC}$ is less than $\kappa_{SVOC}$. It is crucial to include all the effects of SVOCs on activation, including co-condensation because the magnitude of the underestimation of $\kappa_{noCC}$ compared to $\kappa_{SVOC}$ can be of similar magnitude to the overestimation of $\kappa_{nv}$ compared to $\kappa_{SVOC}$.

Semi-volatile organic compounds in the atmosphere give an effective $\kappa$ that is greater than that of the non-volatile compounds, despite the mixing rule causing a reduction in hygroscopicity for many compounds. Consequently, the co-condensation of SVOCs can significantly increase the CCN concentrations observed compared to those that would be expected from the involatile aerosol in environments without SVOCs. There is the necessity for them to be included into large-scale global
models to avoid a drastic underestimation in the number of cloud droplets, with the potential for large global implications if not thoroughly considered.

## Appendix A: Simulating instrument measurements

Aerosol particle sizes in large scale models are often represented by lognormal distributions (Mann et al., 2012) and these are informed by measurements of atmospheric aerosol. The measured size
distributions often contain noise resulting from inaccuracies in the instrument used and the lognormal size distribution is obtained using a best-fit algorithm. The number of lognormal modes used



to represent the measured size distribution will affect the output from the algorithm; specifying too few modes can result in a poor fit but identifying too many can result in fitting an individual mode to large noise artefacts. In the current work we focus only on situations containing a single mode

of particles and in this section we estimate potential uncertainties in the fitting of a lognormal size distribution to noisy measurement data.

Instruments that measure particle sizes measure the number of particles within a certain range of sizes with no information on the distribution of particles with in this range. In addition, inaccuracies in particle sizing can result in some particles being wrongly sized, as well as some particles not being

detected at all. This results in inaccuracies in the size distribution.

In this section, we describe the process of generating random noisy size distribution data from a perfect lognormal size distribution. The following section then describes how this method is incorporated into a Monte-Carlo simulation to generate uncertainties in the number concentration, median diameter and geometric standard deviation of the lognormal size distributions that is used in the main

paper.

We began by specifying a median diameter, geometric standard deviation and number concentration of particles before discretising the size distribution into bins of identically sized particles; this is shown in Figure 11. The black line shows the original lognormal distribution and the bars show the analogous binned quantities. The binned size distribution is used as a proxy for measurement data. In

Sections A1 and A2, we explain how uncertainties in the measured sizes and number concentrations of particles within each bin are simulated.

**A1  Particle sizing uncertainty**

The bars in Figure 11 represent a binned analogy to the lognormal distribution show by the solid black line. The heights of the bars are assumed to be representative of the output from a perfect

instrument that detects and accurately sizes every particle within a lognormal size distribution. In this section, we explain how inaccuracies in the particle sizing are introduced to this idealised case in order to better simulate instrument data.

The particles within a given size bin are assumed to be normally distributed with a mean value equal to the bin centre. The standard deviation of the normal distribution is randomly chosen for each

bin from a uniform distribution within the range 0% and 50% of the bin width. This is inline with the estimated size bin inaccuracies from a Particle Measurement Systems passive cavity aerosol spectrometer probe resulting from the different refractive indices of ammonium sulphate and sulphuric acid (Taylor et al., 2016). Larger standard deviations indicate more inaccuracies in the particle sizing while a standard deviation of zero indicates perfect sizing of a monodisperse aerosol population

within that bin. One such normal distribution is shown by the red line in Figure 11 with the mean indicated by the vertical dashed line. Particles in this bin are redistributed among the neighbouring





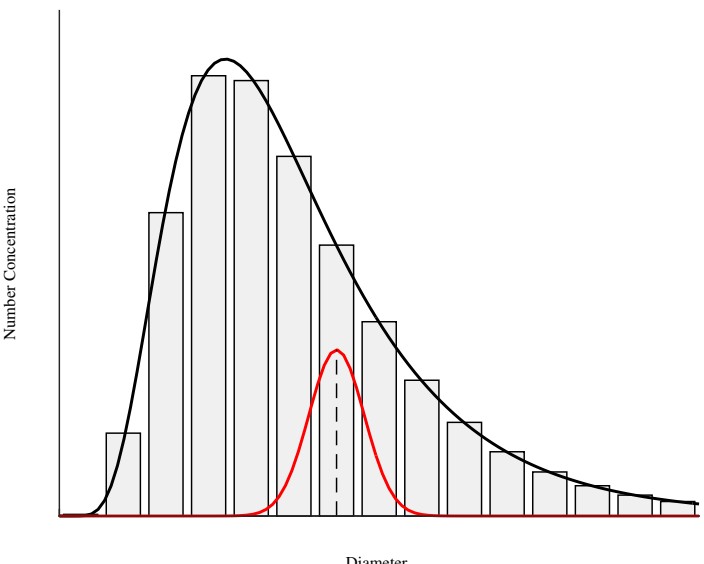

**Figure 11.** A representation of how noisy measurement data is simulated. A lognormal size distribution (black line) is discretised into multiple bins of identically sized particles. The centre of the bar is the size of the particles within each bin. The red line shows the assumed random distribution of particles within a particular size bin.

bins according to this normal distribution. This is carried out for all bins using a different standard deviation for each bin. An example of the resulting binned size distribution is shown in Figure 12.

### A2   Number concentration uncertainty

As well as instruments carrying inaccuracies in particle sizing, instruments that measure particle concentrations can introduce further inaccuracies of approximately 10% (Hering et al., 2005). To simulate this source of uncertainty, we vary the height of each bar in Figure 12 by randomly choosing a number concentration from a normal distribution with mean equal to the number concentration of the bin and a standard deviation of 10% of the mean. The error bars in Figure 12 show one standard

deviation in the number concentration in each bin.

   The total concentration of particles after applying this randomisation remains mostly unchanged due to there being an equal probability of increasing one bin by the same amount as decreasing another. To introduce a variability in total number concentration we subsequently rescale the whole binned size distribution. We randomly choose a new number concentration from a normal distri-

bution with a mean equal to the number concentration of the original lognormal and a standard deviation of $\pm 10\%$ of the mean.





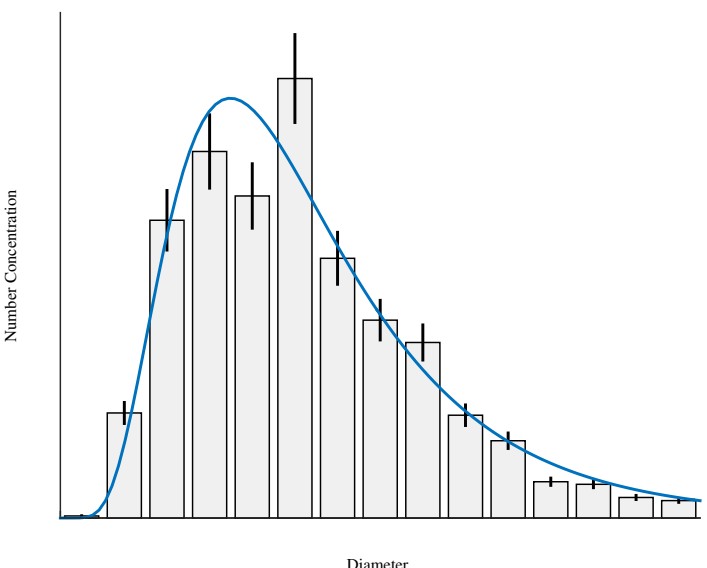

**Figure 12.** The bars in this figure demonstrate the result of randomising the particle sizes within each bin in Figure 11. Uncertainties in the number concentration within each bin are shown by the error bars on the top of each bar, which show one standard deviation. The solid blue line shows the best-fit lognormal size distribution to the randomised binned size distribution.

**A3    Uncertainty in lognormal size distribution parameters**

In this section we demonstrate how randomising the binned size distributions translates into uncertainties in the fitted lognormal size distributions that are used in the main paper. A schematic of this process is shown in Figure 13. For a given number concentration, $N$, median diameter, $D_m$, and geometric standard deviation, $\ln \sigma$, the corresponding lognormal size distribution was binned and random noise added as described in Section A. A new lognormal size distribution was fitted to the resulting bar chart size distribution to find new values of the number concentration, $N_{fit}$, median diameter, $D_{fit}$, and geometric standard deviation, $\ln \sigma_{fit}$. This process was repeated multiple times and the mean and standard deviation for each parameter value calculated, which are denoted by superposed bars and hats, respectively. The whole process was repeated for the different size distribution parameters chosen randomly from uniform distributions with maxima and minima given in table Table 6.

Figure 14 shows the uncertainty in the median diameter of the fitted lognormal size distributions for a range of median diameters of the initial lognormal mode, $D_m$. For each initial value of the median diameter, $D_m$, the mean $\bar{D}_{fit}$ and standard deviation $\hat{D}_{fit}$ of median diameters of the fitted lognormal modes are shown by the black dots and blue vertical lines, respectively. The 1:1 line is





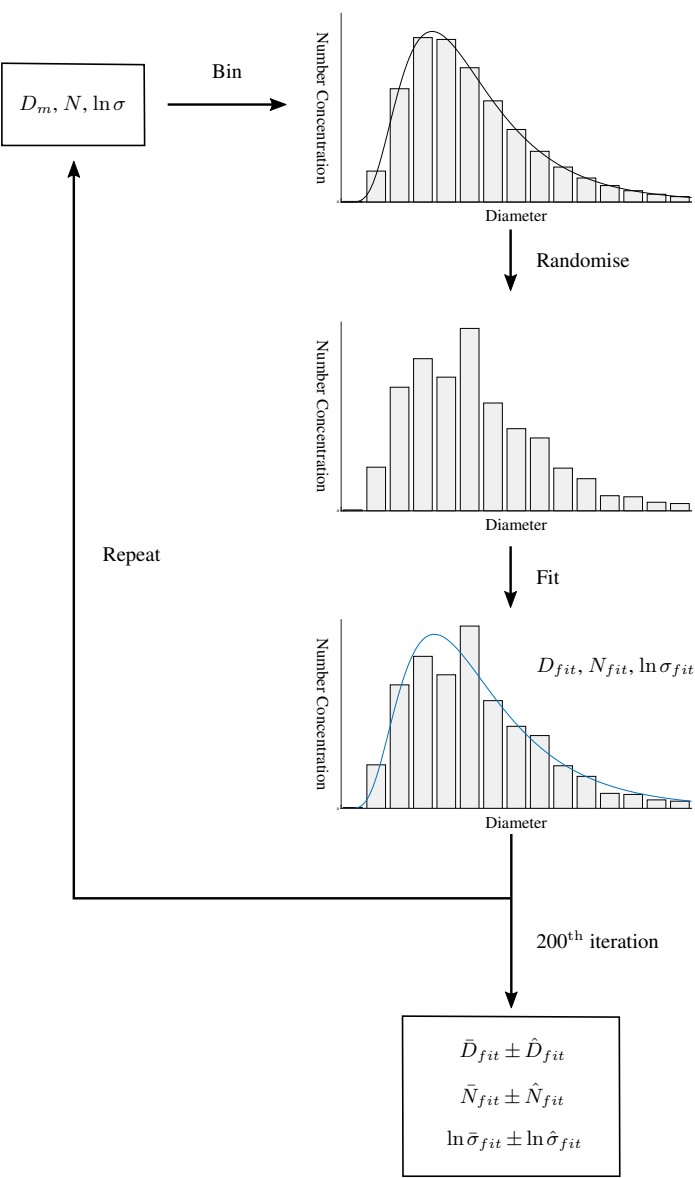

**Figure 13.** Schematic showing how the uncertainty in the size distribution parameters is derived using the randomisation process in Section A. Initial size distribution parameters, $D_m$, $N$, and $\ln\sigma$ are chosen and the size distribution binned into 20 nm wide size bins. The concentration in each bin is then randomised and a new lognormal size distribution fitted to the bar bar. After repeating 200 times, the mean and standard deviation in each of the parameters of the fitted size distributions are calculated. This process is repeated multiple times for different values of $N$, $D_m$ and $\ln\sigma$ taken from Table 6.





**Table 6.** Minimum and maximum of the size distribution parameters used for Figures 14 to 16.

| Parameter | minimum | maximum |
|---|---|---|
| $N$ (cm$^{-3}$) | 100 | 2000 |
| $D_m$ (nm) | 50 | 500 |
| $\ln \sigma$ | 0.1 | 1 |

shown by the solid black line and lines showing deviation from the 1:1 line of 5% and 10% are shown by the dashed, and dot-dashed lines. In all cases, the number concentration was between 100 cm$^{-3}$

and 2000 cm$^{-3}$ and randomly chosen and the geometric standard deviation was randomly selected from the range between 0.1 and 1. The mean values of the fitted median diameters deviate very little from the 1:1. The blue vertical error bars show deviation from the mean of one standard deviation, namely $\bar{D}_{fit} \pm \hat{D}_{fit}$, and for all initial median diameter values the error bars mostly lie within the dot-dashed lines showing 10% deviation from the mean. Therefore, we represent the uncertainty in

the median diameter of the non-volatile modes by a standard deviation of 10% of the mean.

Figure 15 shows the uncertainty in the number concentration of the fitted lognormal size distributions. The black dots show the mean number concentration of the fitted lognormal modes, $\bar{N}_{fit}$, against the original value of $N$. In the majority of cases, there is less than 5% discrepancy between $\bar{N}_{fit}$ and $N$. The uncertainty in the fitted number concentration is shown by the vertical blue lines

that represent $\bar{N}_{fit} \pm \hat{N}_{fit}$. In most cases, the uncertainty in the fitted number concentration deviates from the mean by about 10% - 15%. There are some cases that have error bars extending beyond the dotted lines, which show 15% deviation from the 1:1 line, however, these mostly correspond to the cases where $\bar{N}_{fit}$ deviates from $N$ by 10% - 15%. Consequently, the associated error bars still show a standard deviation of 10% of $\bar{N}_{fit}$.

Figure 16 shows the mean and standard deviation in the geometric standard deviation of the fitted lognormal modes, $\ln \sigma_{fit}$. A distinct feature of this figure that was not present in Figures 14 and 15 is that the mean of the fitted geometric standard deviations, $\ln \bar{\sigma}_{fit}$, do not agree with the initial values of $\ln \sigma$ but is approximately 0.1 higher. This is an artefact of the randomisation process that moves particles between each bin. Particles are passed from one size bin into the neighbouring bins

and these bins pass particles back again. On average, size bins that contain the most particles will pass more particles into the neighbouring bins that contain fewer particles than are passed back. As a result, the particles in the size bins containing the most number of particles diffuse out into the smaller and larger diameter size bins. This produces a wider size distribution and, therefore, larger values of $\ln \bar{\sigma}_{fit}$ compared to $\ln \sigma$. The purpose of this graph, however, is to ascertain the relative

size of the standard deviation, $\ln \hat{\sigma}_{fit}$, compared to the mean value $\ln \bar{\sigma}_{fit}$. For all values of $\ln \sigma$ the error bars, which show $\ln \bar{\sigma}_{fit} \pm \ln \hat{\sigma}_{fit}$, span approximately 0.1 from the mean. As such, we use an uncertainty in the geometric standard deviation of 0.1.

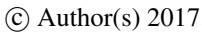



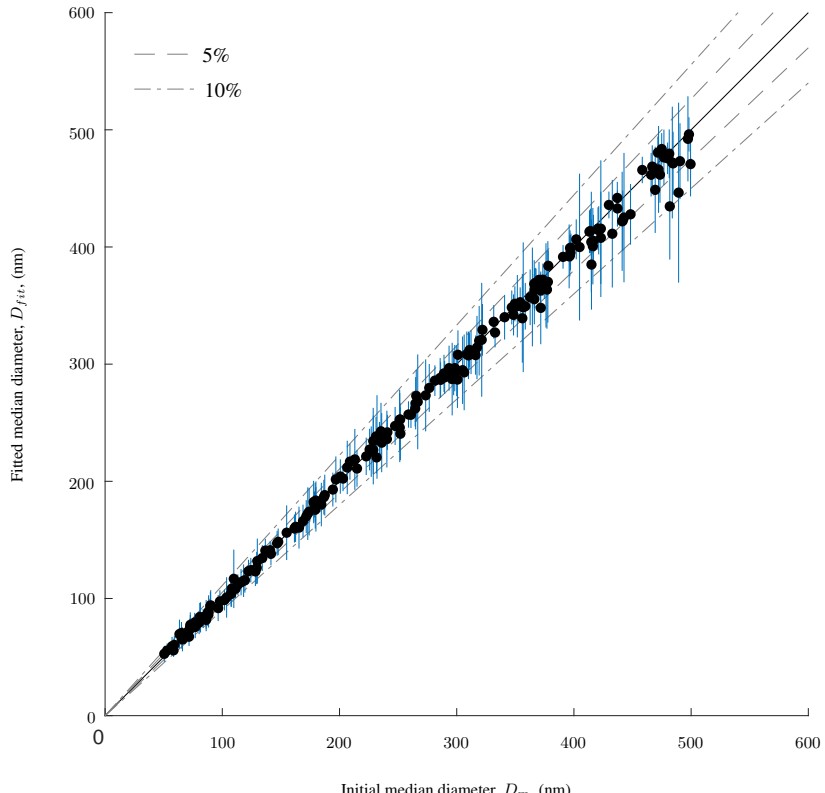

**Figure 14.** Uncertainty ranges in the median diameter of the fitted lognormal size distribution parameters that result from the randomisation process in the aerosol size distribution. The 1:1 line between $D_{fit}$ and $D_m$ is shown by the solid line and the dashed and dot-dashed line show deviation of 5% and 10% from the 1:1 line respectively. Black dots show the mean of the fitted median diameters, $\bar{D}_{fit}$, and the standard deviation, $\hat{D}_m$, is shown by the vertical error bars.

## Appendix B: SVOC uncertainty

### B1 Volatility distribution uncertainty

The volatility distribution of SVOCs in the atmosphere is the largest source of uncertainties discussed in this paper. Not only is the total concentration of SVOCs uncertain but also the volatility of the different compounds. Due to the volatile nature of the SVOCs, making accurate measurements is very difficult, often resulting in several orders of magnitude variability in the saturation vapour pressure of each individual compound.

In this paper, we use a volatility basis set and assume the distribution is that measured by Cappa and Jimenez (2010) but we add a random element to simulate the uncertainty. This is achieved in



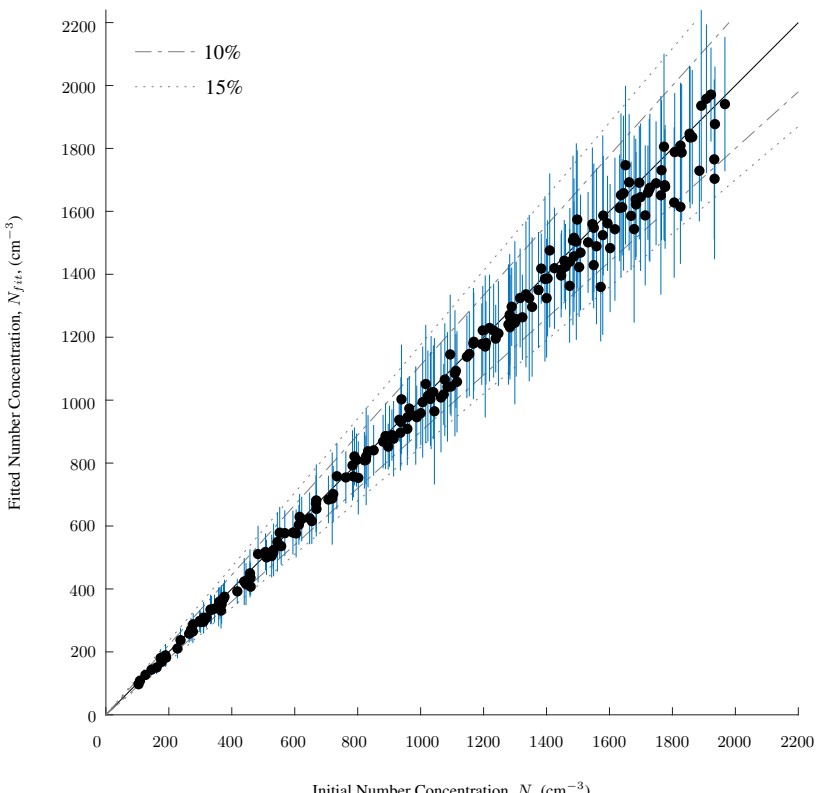

**Figure 15.** Same as Figure 14 for the number concentration of the fitted lognormal size distributions. Mean values and standard deviations of the fitted number concentrations are shown by the black dots and error bars, respectively.

a similar way to the size distribution. A schematic is shown in Figure 17. The unaltered volatility distribution is shown in the top left of the Figure. The volatility of the compounds represented by the dark green bar are assumed to be normally distributed, rather than all having a single $C^*$ value,

and this simulates uncertainties in measuring the saturation vapour pressures. For each volatility bin, the mean is taken to be the $C^*$ value of the volatility bin and the standard deviation of the normal distribution is chosen randomly from a uniform distribution ranging from 0% to 33% of the mean $\log_{10} C^*$ value, based on the measured compounds in Bilde et al. (2015). The normal distribution is then integrated between each volatility bin boundary and the corresponding mass redistributed

between neighbouring bins. This process is repeated for each volatility bin using the mass in the original volatility distribution. A new randomised volatility distribution is then created by adding together all of the binned normal distributions. We then simulate uncertainties in measuring the organic mass in each volatility bin. To do this we assume the mass in each volatility bin is normally




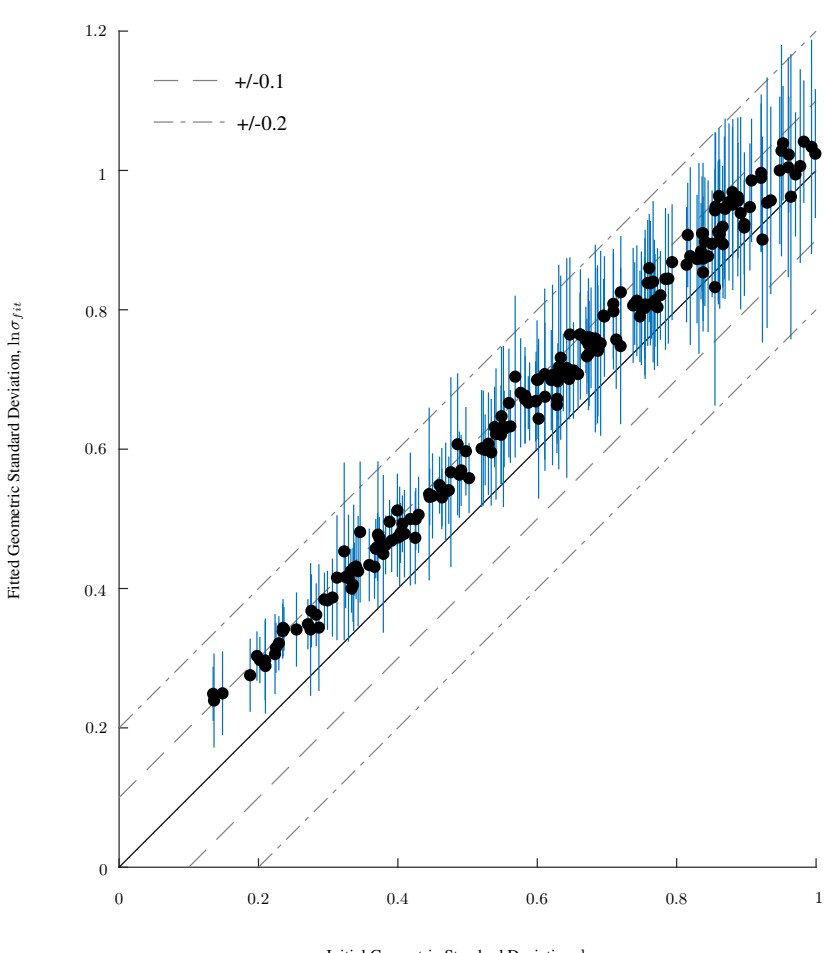

**Figure 16.** Same as Figure 14 for the geometric standard deviation of the fitted lognormal size distributions. Mean values and standard deviations of the fitted geometric standard deviation are shown by the black dots and error bars, respectively.





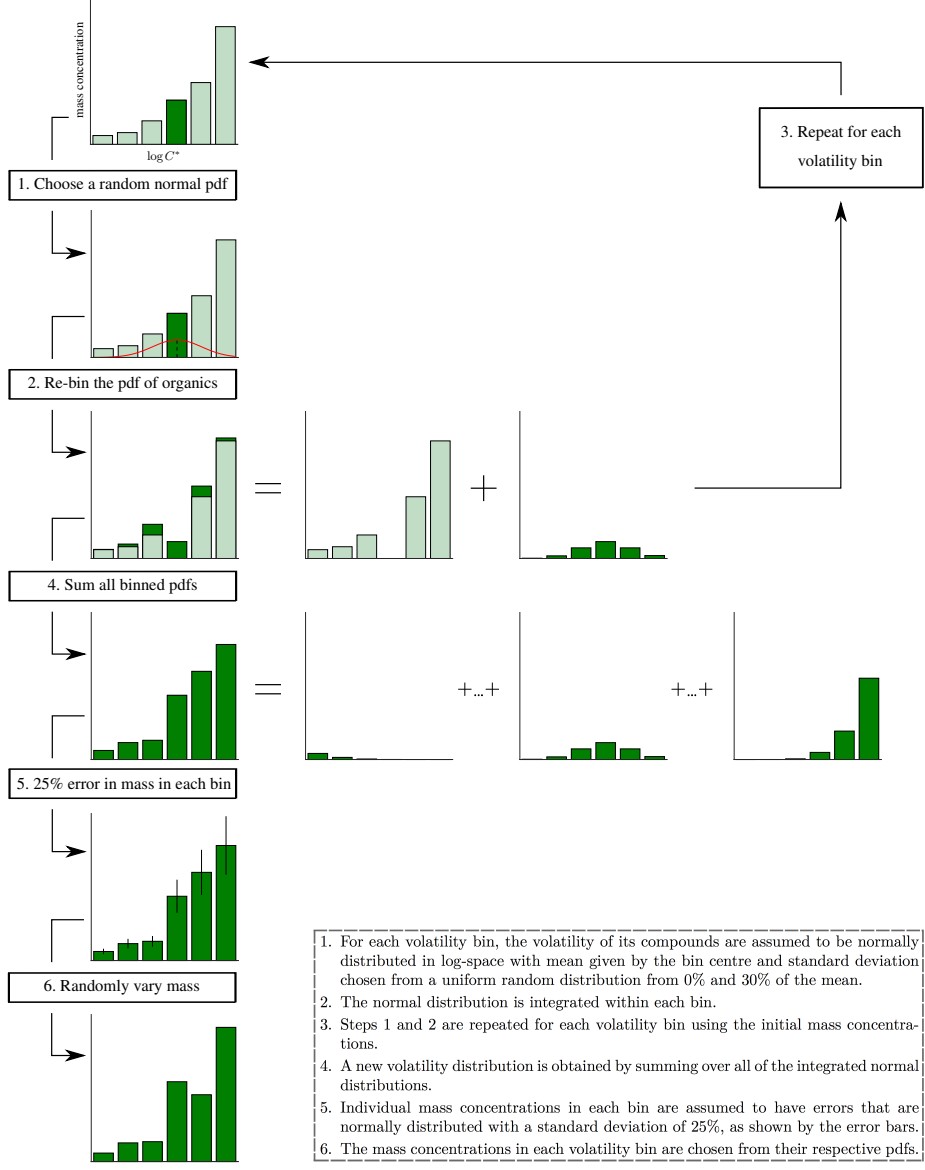

1. For each volatility bin, the volatility of its compounds are assumed to be normally distributed in log-space with mean given by the bin centre and standard deviation chosen from a uniform random distribution from 0% and 30% of the mean.
2. The normal distribution is integrated within each bin.
3. Steps 1 and 2 are repeated for each volatility bin using the initial mass concentrations.
4. A new volatility distribution is obtained by summing over all of the integrated normal distributions.
5. Individual mass concentrations in each bin are assumed to have errors that are normally distributed with a standard deviation of 25%, as shown by the error bars.
6. The mass concentrations in each volatility bin are chosen from their respective pdfs.

**Figure 17.** A schematic showing the method used to randomly vary the volatility distribution given in Table 4. The final distribution is rescaled so that the total mass concentration results in a specified organic mass fraction in the aerosol size distribution at 50% RH. In each simulation, organic mass fraction is chosen randomly from a uniform distribution between 10% and 50%.





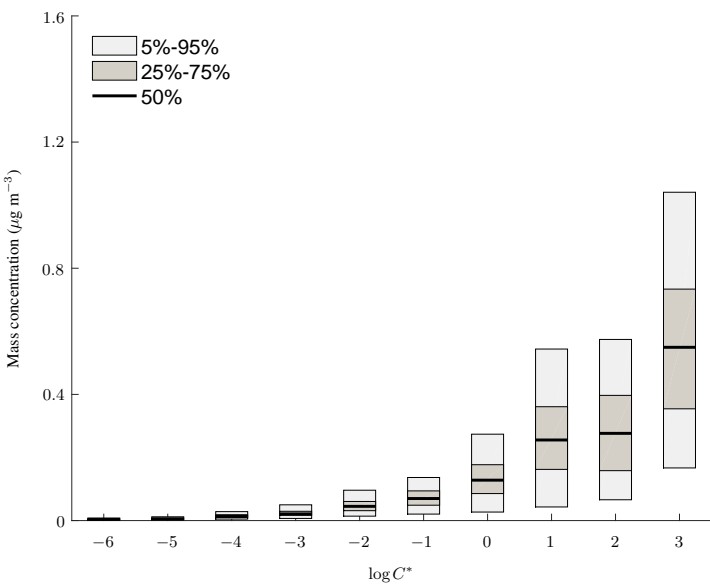

**Figure 18.** Example of the variability of mass concentration in each volatility bin. The 5% to 90% quantiles are shown by the lighter grey bars, and the interquartile range is shown by the darker shaded area. The median mass concentrations are shown by the horizontal black lines.

distributed with mean obtained from step 4 and standard deviation of 25% of the mean. New masses are then chosen from the resulting normal distributions. All SVOCs with a $C^*$ value less than the lowest volatility bin are placed in the first bin and, similarly, all SVOCs with a higher $C^*$ value than the highest volatility bin are placed in the last bin.

For a given volatility distribution with total mass loading of $1$ $\mu g$ $m^{-3}$, Figure 18 shows the total variability of the mass concentration in each bin. A Monte Carlo simulation was run 100 times with parameters randomly chosen as described above. Different quantiles are plotted to show the variability in the mass concentration in each bin.

### B2   Material parameter uncertainty

Table 7 shows the molecular weight and density of a variety of compounds found in biogenic SOA. The molecular weights are approximately uniformly distributed. A volatility basis set approach involves binning large numbers of different compounds with different mass concentrations and material properties. For each of our simulations, we simulate volatility basis sets that are composed of random numbers of compounds in different ratios. Although the molecular weights of individual compounds are uniformly distributed, the result of combining multiple random compounds and averaging their individual molecular weights will result in a range of possible effective molecular





weights that are normally distributed. This is because choosing a compound with a large molecular
weight is likely to be mitigated by also choosing a compound with a very small molecular weight.
As a result, we choose molecular weights from a normal distribution with mean 200 g mol$^{-1}$ and
standard deviation of 100 g mol$^{-1}$. We further restrict the values to lie between 100 g mol$^{-1}$ and
300 g mol$^{-1}$ to avoid unphysical values. A similar approach is used for the van't Hoff factors and

the densities and are summarised in Table 5.

**Table 7.** Table of molecular weight and density for a range of biogenic SOA.

| Compound | Molecular Weight (g mol$^{-1}$) | Density (g mL$^{-1}$) |
|---|---|---|
| $\alpha$-Pinene oxide | 152 | 0.964 |
| $\beta$-Pinene oxide | 152 | 0.97 |
| Limonene-2-oxide | 152 | - |
| d-Limonene | 136 | 0.8411 |
| 2-Hydroxy-3-pinanone | 168 | - |
| Cineole | 154 | 0.922 |
| Myrtenal | 150 | 0.988 |
| Citronellal | 154 | 0.855 |
| Citral | 152 | 0.893 |
| Ketopinic acid | 182 | 1.238 |
| *cis*-Isoketopinic acid | 182 | - |
| Nopinone | 138 | 0.981 |
| Menthone | 154 | 0.895 |
| Camphore | 152 | 0.99 |
| Myrtenol | 152 | 0.954 |
| *cis*-Verbenol | 152 | - |
| *trans*-Pinane-1,10-diol | 170 | - |
| *trans*-p-Menth-6-en-2,8-diol | 170 | - |
| Pinanediol | 170 | - |
| Bornyl acetate | 196 | 0.986 |
| Geranyl acetate | 196 | 0.916 |
| Linalyl acetate | 196 | 0.895 |

*Acknowledgements.* The research leading to these results has received funding from NERC, through the Re-
search Experience Placement (REP) scheme, and the European Union's Seventh Framework Programme (FP7/2007-
2013), under grant agreement n° 603445.




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
