# Peer review of "Uncertainty in aerosol hygroscopicity resulting from semi-volatile organic compounds"

_Atmospheric Chemistry and Physics, 2017_

## Referee Comment (RC1) · Anonymous Referee #1 · 22 May 2017

Aerosol hygroscopicity plays a major role in determining the ability of aerosol acting as cloud condensation nuclei (CCN), which has an important effect on cloud and thus climate. Most of current existing cloud droplet activation parameterizations neglect the effect of organic compounds on hygroscopicity, despite of the large amount of organic aerosols in the atmosphere.

Goulden et al. (2017) paper examined the effect of semi-volatile organic compounds (SVOCs) on aerosol hygroscopicity and quantified its uncertainties. The paper concluded that the including SVOCs tend to decrease the aerosol hygroscopicity primarily because of the lower hygroscopicity of SVOCs than those of non-volatile aerosols. The paper also proposed a parameter, called effective hygroscopicity, to account for the effect of SVOCs on cloud droplet number concentrations. The effective hygroscopicity was shown to be higher than the original hygroscopicity of non-volatile aerosols.

[Figure]

For sure, this paper is of great interest to the aerosol/climate community and will help us understand the effect of organic aerosol on cloud formation. Yet, it needs improvement to better present its results and to clarify its ambiguous results.

Major comments

One pressing issue is that results shown in the paper seems inconsistent. The paper first showed that the inclusion of SVOCs leads to a lower hygroscopicity (kappa), which implies that neglecting SVOCs would overestimate kappa, and thus CCN number as well. However, Âăthe paper later showed the effective kappa is higher when including SVOCs .This means that neglecting SVOCs would underestimate kappa, and also CCN number. These two conclusions seems contrast each other. Can the authors clarify this?

The paper was not well structured, with many sudden jumps between paragraphs and sections, causing me a lot of trouble to follow. Here are some of my suggestions to improve that.

- Page 3, last paragraph. The introduction of the three single-parameter measures of the hygroscopicity seems abrupt, causing confusion without further explanations. I suggest placing this introduction in the beginning of section 4: methodology including the effects of SVOCs.

- Before moving to the main body of paper, please briefly lay out the structure of the following content, telling readers what they would expect in the coming paper.

- Reorganize the main body of the paper. The current section form starts from methodology, then to results, and jump back to methodology and results, which I think is not fluent. Two ways to fix it. 1) put all methodology parts into one section, followed by the results section; 2) Combine the section 2 and section 3 into one part as for involatile aerosol with section 2 and 3 as sub-sections, and combine the section 4 and 5 into the other part as for including the effects of SVOCs.

Minor comments

Page 2, line 28. How about the recent IPCC results (AR5)?

Page 3, line 69-72. What did the author mean by 'dynamic condensation'? That is inconsistent with 'equilibrium absorptive partitioning' stated in the beginning of this sentence.

Page 3, line 91. According to the equilibrium absorptive partitioning theory, the primary factors controlling the gas/particle partitioning are the vapor pressure of SVOCs, atmospheric temperature, and the total mass of existing particles, without RH, although RH is relative to the temperature. Can the authors explain more why they particularly chose RH?

Page 4, line 109-111. This sentence seems odd to me. Âă"Many source of uncertainties" in the first part is logically disconnected to the second part of this sentence.

Page 5, line 166. A little confusion here. How did the authors obtain the "12%" value?

Page 8, Line 210. "Table 1". Did the authors mean Table 3?

Page 11, Line 268. Any specific reason that 50% of RH was chosen for the integration of aerosol size distribution? Not 60%? Any effect on the derived hygroscopicity if using different RH?

Page 11, line 287. Why does it have to be between 0.1 and 0.5?

Page 12, line 291. Can the authors remind the readers what the parameterization of Connolly et al. (2014) is? Since it is first introduced in the Introduction Section, which is quite far away from here.

Page 13, line 313-315. The smaller uncertainty for k_SVOC than k_nv is quite surprising, because the uncertainty for k_SVOC includes the uncertainty associated with not only the non-volatile particles but also SVOCs volatilities and masteries, while the uncertainty for k_nv reflects only the non-volatile particles related uncertainty. Do the

authors know why?

Page 14, line 325. "50% RH" is different from "0%RH" stated on line 323. Which one is right?

Page 16, line 368. Shouldn't it be wet aerosol size distribution, because at 70% RH, for example, the aerosol can absorb water?

Figure 7. What are the red + in the top of figure?

Figure 8. Why the line of 100% shows a different trend than other lines at high logC bins?

Section 6. As a large portion of the paper concentrates on the uncertainty of hygroscopicity associated with involatile aerosol size distribution and SVOCs mass and chemical compositions, I think the authors should add the findings about this uncertainty part, which can also echo the title of the paper.

Section 6. The results are derived from the assumption that SOA is the result of the equilibrium absorptive partitioning of SVOCs, but some experimental results indicate that aerosol particles containing SOA can exist in highly viscous states (e.g., Vaden et al., 2011 PNAS), breaking the equilibrium partitioning. Would the viscous states of particles change the results of this paper?

---

## Short Comment (SC1) · 22 May 2017

The manuscript "Uncertainty in aerosol hygroscopicity resulting from semi-volatile organic compounds" by Goulden, et al. is quite long and difficult to read, in my opinion. A complete rewrite with an eye toward tightening (an overall decrease of 25% seems possible and desirable) and clarity of language and logical construction would probably greatly improve the readability. The paper conducts a sensitivity analysis (a relatively mature mathematical method) of an existing set of parameterizations for cloud droplet formation. The core is a standard non-volatile aerosol activation model by Nenes which the authors have extended to allow co-condensation of SVOCs resulting in an apparent increase in the hygroscopicity of the "dry" particles (in some cases because the core particle is made more hygroscopic by the inclusion of the SVOC and in all cases – even those where the particle is made less "water loving" – because the particles are larger

[Figure]

Interactive
comment

during the cloud updraft). In the interest of full disclosure, I am more of an experimentalist than a modeler, but I think the work on the parameterizations is very important and interesting, while this treatment of the aggregate uncertainties of the model is less so.

Some specific areas that I would focus on, if it is determined that a major rewrite is needed are: 1) As noted above, the concept of varying input parameters over likely ranges and determining the sensitivity of the resultant products (Smax, Dmin, kappa) to this is not foreign to most readers, so a terse explanation and tabulation of the ranges used would probably suffice. I would particularly recommend minimizing the discussion surrounding the core model (Sec. 3), where few new physical insights were produced. 2) A separate discussion of the modeling (Fig. 3 and surrounding text) with and without SVOC effects seems unwarranted – I would submit that a more concise discussion of the full implementation that notes the logical intermediate "off ramps" explored in this paper would probably be easier for most readers, even those unfamiliar with the concept. 3) In my opinion, there is too much discussion of the intermediate test cases (e.g., Knocc) and much of it is presented in an odd "event drives cause" manner that I found pervasive throughout the manuscript, for instance "For levoglucosan, the mixing rule has a less dominant effect than the increase in size at cloud base ...." 3a) The presentation of the levoglucosan results should be strongly caveated, since the results are apparently contradictory to the general thrust of the paper. Clearly this is an extreme case where a very hygroscopic core is exposed to a relatively less hygroscopic SVOC and the final product is still an apparently easy to activate particle. 4) Finally, and probably most importantly, I would recommend more/clearer discussion of the proposed use of an "effective hygroscopicity" in parameterizations used in larger scale models. It appears to me that the authors recommend simply "adjusting" the hygroscopicity of well-characterized particle types upward to account for the SVOC/water co-condensation, apparently without regard for the amount or nature of the SVOC that the aerosols are likely to have been exposed to. In my opinion, this makes as little sense as not accounting for the co-condensation in extant models and will probably

result in a significant overestimation of the cloud formation and importantly also the sub-critical water uptake, resulting in a distortion of the optical properties. If this isn't what the authors are suggesting, I believe they should clarify this point.

Because the subject matter of this manuscript is of clear importance (although I do not think the work here is central to that effort, as it seems to be offering little new physical insight) I would think it is publishable. But I highly recommend an effort at recrafting it to make it a tighter, easier to read paper.

---

## Author Comment (AC1) · 21 Aug 2017

Q: One pressing issue is that results shown in the paper seems inconsistent. The paper first showed that the inclusion of SVOCs leads to a lower hygroscopicity (kappa), which implies that neglecting SVOCs would overestimate kappa, and thus CCN number as well. However, the paper later showed the effective kappa is higher when including SVOCs.This means that neglecting SVOCs would underestimate kappa, and also CCN number. These two conclusions seems contrast each other. Can the authors clarify this? A: The inclusion of SVOCs increases the size of particles as well as changing the chemical composition and so the hygroscopicity has a more complicated relationship to CCN concentration as would be the case for involatile particles. Ignoring composition, larger particles activate at lower supersaturations and one may expect more CCN as a

result of the increased size of particles with the SVOCs. The hygroscopicity, however, is also inversely proportional to the critical diameter cubed (equation 1). Hence, if the particles that activate are larger, the resulting hygroscopicity is smaller.

Q: Page 3, last paragraph. The introduction of the three single-parameter measures of the hygroscopicity seems abrupt, causing confusion without further explanations. I suggest placing this introduction in the beginning of section 4: methodology including the effects of SVOCs A: This paragraph has been moved to the methodology section, as suggested.

Q: Before moving to the main body of paper, please briefly lay out the structure of the following content, telling readers what they would expect in the coming paper. A: This has been added at the start of the final paragraph of the introduction.

Q: Reorganize the main body of the paper. The current section form starts from methodology, then to results, and jump back to methodology and results, which I think is not fluent. Two ways to fix it. 1) put all methodology parts into one section, followed by the results section; 2) Combine the section 2 and section 3 into one part as for involatile aerosol with section 2 and 3 as sub-sections, and combine the section 4 and 5 into the other part as for including the effects of SVOCs. A: The sections on the involatile aerosol (method and results) was presented first to allow the reader to understand the Monte-Carlo methodology when applied to the simpler and familiar definition of hygroscopicity. In addition, experimental results are available for comparison in this case. We the go on to describe the addition of the SVOCs and discuss that this more complicated due to the variation in size and chemical composition with RH. There are no experimental results for comparison and so this builds on the current theory and understanding.

Q: Page 2, line 28. How about the recent IPCC results (AR5)? A: Citation has been updated to Myhre et al. (2013)

Q: Page 3, line 69-72. What did the author mean by 'dynamic condensation'? That

is inconsistent with 'equilibrium absorptive partitioning' stated in the beginning of this sentence. A: In dynamic parcel model simulations by Connolly et al. (2014), it was found that the SVOCs approximately reached equilibrium between the vapour and total condensed phase by cloud base. The condensed mass on each individual particle within the polydisperse aerosol, however, was not in equilibrium. Consequently, the total condensed mass was calculated using equilibrium absorptive partitioning theory but this mass is distributed between the different particle sizes within a lognormal mode by changing the median diameter and geometric standard deviation. This change in lognormal parameters is used to "simulate the condensed phase of SVOCs after undergoing dynamic condensation".

Q: Page 3, line 91. According to the equilibrium absorptive partitioning theory, the primary factors controlling the gas/particle partitioning are the vapor pressure of SVOCs, atmospheric temperature, and the total mass of existing particles, without RH, although RH is relative to the temperature. Can the authors explain more why they particularly chose RH? A: We have used the equilibrium absorptive partitioning model of Barley et al. (2006) that includes condensed water in the mass of the particle phase. The mass of water is controlled by the RH. Near 100% RH the mass of water in the particle phase increases dramatically and this induces signififcant condensation of SVOCs.

Q: Page 4, line 109-111. This sentence seems odd to me. "Many source of uncertainties" in the first part is logically disconnected to the second part of this sentence. A: The sentence is trying to say that we're encapsulating the many sources of uncertainty into a single parameter with associated uncertainty. This has been reworded to make it clearer.

Q: Page 5, line 166. A little confusion here. How did the authors obtain the "12%" value? A: The 12% resulted from the method of randomising the size distribution described in the Appendix A (now in the supplement). In the interest of reducing the length of the paper, as requested by both reviewers, this paragraph has been removed

Q: Page 8, Line 210. "Table 1". Did the authors mean Table 3? A: Yes, this has been corrected in the text.

Q: Page 11, Line 268. Any specific reason that 50% of RH was chosen for the integration of aerosol size distribution? Not 60%? Any effect on the derived hygroscopicity if using different RH? A: Air is often dried to 50% RH before measuring the aerosol size distribution. Figure 7 (Now Figure S7 in the Supplement) shows the effects of using higher RH values with little difference in the hygroscopicity below 95% RH.

Q: Page 11, line 287. Why does it have to be between 0.1 and 0.5? A: These are typical ranges measured for organic fraction of aerosol.

Q: Page 12, line 291. Can the authors remind the readers what the parameterization of Connolly et al. (2014) is? Since it is first introduced in the Introduction Section, which is quite far away from here. A: It is not that easy to briefly describe the parameterisation and a major criticism of the paper was its length. Adequate references to the original parameterisation are included.

Q: Page 13, line 313-315. The smaller uncertainty for k_SVOC than k_nv is quite surprising, because the uncertainty for k_SVOC includes the uncertainty associated with not only the non-volatile particles but also SVOCs volatilities and masteries, while the uncertainty for k_nv reflects only the non-volatile particles related uncertainty. Do the authors know why? A: The 10% is for levoglucosan while the other 3 compounds the difference is more like 5%, the text has been changed to reflect this. For ammonium sulphate, sodium chloride and sulphuric acid the hygroscopicity is large and small changes in the size of the particles can make a big difference to the number of CCN. The organics, however, have a hygroscopicity of between 0.18 and 0.27 (based on Table 5), which is "low" regardless of what value it actually takes in this range. Overall, the combination of the involatile constituents and the organic compounds will result in relatively low hygroscopic particles regardless of the particular properties of the organics. Furthermore, the uncertainty in organic mass fraction will have a large effect on

the chemical composition of the particles, however, this is mitigated by the changes in sizes of the particles due to the SVOCs.

Q: Page 14, line 325. "50% RH" is different from "0%RH" stated on line 323. Which one is right? A: "0%" is a typo and has been corrected to 50%

Q: Page 16, line 368. Shouldn't it be wet aerosol size distribution, because at 70% RH, for example, the aerosol can absorb water? A: The equation for the hygroscopicity, (1), depends on the dry particle size, as defined in Petters and Kreidenweis (2007). In the case with SVOCs, the dry size neglects the condensed water but includes the associated condensed SVOCs at the particular RH.

Q: Figure 7. What are the red + in the top of figure? A: These are the data point that MATLAB deems outliers and are the points that are more than 1.5 times the interquartile range below and above the 25th and 75th percentiles, respectively.

Q: Figure 8. Why the line of 100% shows a different trend than other lines at high logC bins? A: The partitioning of the SVOCs into the condensed phase depends on the condensed water. At higher RH there is more condensed water and therefore more SVOCs in the condensed phase. The is significantly more water condensed at 100% RH than below 95% RH and so the total condensed mass of SVOCs will be significantly higher. By 95% humidity, the organics in the lower 8 volatility bins have all partitioned into the particle phase and so the additional condensed SVOC mass at 100% RH must come from the higher 2 volatility bins.

Q: Section 6. As a large portion of the paper concentrates on the uncertainty of hygroscopicity associated with involatile aerosol size distribution and SVOCs mass and chemical compositions, I think the authors should add the findings about this uncertainty part, which can also echo the title of the paper. A: The focus of this paper on the effect of SVOCs. The involatile section (Section 2) is included to allow some comparison with existing experimental data that, in comparison to when SVOCs are included, is more straight forward to measure. We are therefore presenting a method of including

SVOCs that does not have the pit-falls associated with measuring SVOCs. Additionally, reviewer 2 felt there was already too much focus on the involatile case. (Point 1)

Q: Section 6. The results are derived from the assumption that SOA is the result of the equilibrium absorptive partitioning of SVOCs, but some experimental results indicate that aerosol particles containing SOA can exist in highly viscous states (e.g., Vaden et al., 2011 PNAS), breaking the equilibrium partitioning. Would the viscous states of particles change the results of this paper? A: This is true but is an active area of research. Some numerical models investigating the effect of diffusion within aerosol particles (Zobrist et al., 2011; Smith et al., 2003) indicate that glassy particles can transition into a liquid phase above about 50% relative humidity. As this paper focuses on relative humidities above 50% the effect of viscosity may not be important.

---

## Author Comment (AC2) · 21 Aug 2017

1) "As noted above, the concept of varying input parameters over likely ranges and determining the sensitivity of the resultant products (Smax, Dmin, kappa) to this is not foreign to most readers, so a terse explanation and tabulation of the ranges used would probably suffice. I would particularly recommend minimizing the discussion surrounding the core model (Sec. 3), where few new physical insights were produced." A: The length of the paper has been reduced and hopefully addresses this point.

2) "I would particularly recommend minimizing the discussion surrounding the core model (Sec. 3), where few new physical insights were produced" A: We disagree that "few new physical insights were produced). The purpose of the involatile aerosol section is to try to demonstrate that the uncertainty in our model is similar to that in experiments with a view to justifying the suitability of our model to represent real-world problems. This is surely important for an model. Additionally, it is found that the uncertainty in kappa when SVOCs are included is smaller than that of the involatile aerosol. This is, perhaps, an unexpected observation as noted by Reviewer 1 ("Page 13, line 313-315") and can only be drawn if the involatile aerosol is studied in isolation. Hopefully the more concise wording of this section in response to the reviewers' comments has addressed this issue.

3) there is too much discussion of the intermediate test cases (e.g. KnoCC) A: Large-scale models that include VOCs often only consider the condensed phase of the organics under subsaturated conditions (REF). These models effectively work with the KnoCC parameter, which only takes into account the condensed phase of organics at subsaturated conditions. KnoCC is not, therefore, an intermediate step but an important parameter in the discussion of how well current models may be representing the effect of SVOCs on cloud. An additional paragraph has been added to the introduction (lines 94-104) that discusses challenges and methods involved in different experimental and modelling methods.

3a) The presentation of the levoglucosan results should be strongly caveated, since the results are apparently contradictory to the general thrust of the paper. A: We do not believe the results for levoglucosan are contradictory. The effect of the SVOCs is clearly dependent on the composition of the involatile aerosol and the results for levoglucosan show this: producing a higher kappa with SVOCs than without while the other 3 compounds have lower kappa with SVOCs than without. Q: Clearly this is an extreme case where a very hygroscopic core is exposed to a relatively less hygroscopic SVOC and the final product is still an apparently easy to activate particle. A: We, also, do not believe that levoglucosan is an extreme case; it is quite a common compound to use in models and its hygroscopicity is not outside of the range of typical kappa values that have been measured for other compounds (See Petters and Kreidenweis, 2007).

It is also fairly hydrophobic, not "very hygroscopic", we assume this is just a typo.

4) It appears to me that the authors recommend simply "adjusting" the hygroscopicity of well-characterized particle types upward to account for the SVOC/water co-condensation A: We are suggesting that the hygroscopicity can be adjusted to account for the co-condensation of SVOCs to produce similar numbers of CCN. We are not, however, suggesting that this be done "without regard for the amount or nature of the SVOC that the aerosols are likely to have been exposed to". The results in this paper are only to illustrate typical trends in the hygroscopicity that might be expected and we do not suggest that the hygroscopicity of sodium chloride, for example, be replaced with a value of 2.1. Our method could be repeated with whatever SVOC properties and abundance is used in a particular model. Specifically, our method would be useful for models such as GLOMAP (Mann et al., 2010) that have no capacity to treat SVOCs. To include their effects would involve running multiple large-scale models to redefine the parameters used in GLOMAP. Our method, in contrast, could be carried out off-line and the modified hygroscopicities used in the existing GLOMAP model instead.

Mann, G.W. et al., 2010. Description and evaluation of GLOMAP-mode: a modal global aerosol microphysics model for the UKCA composition-climate model. Geoscientific Model Development, 3(2), pp.519–551. Available at: http://www.geosci-model-dev.net/3/519/2010/.

---

## Author Response (AR2)

**Response to second review by Referee #2**
The suggest improvements to schematic figure were very helpful and have been implemented. We feel that the information in Table 1, however, are numbers that a reader may wish have made obvious as no comprehensive study on the effect of varying these parameters has been carried out in the main paper (except for the few test choices in the supplement). Additionally, the information in Table 2 is more concisely displayed through a table. We feel that to write the same information in text would be quite cumbersome and would not improve the readability of the paper.

[revised manuscript text omitted]